# Filtered Inner Product Projection for Crosslingual Embedding Alignment

**Vin Sachidananda**
Stanford University
vsachi@stanford.edu

**Ziyi Yang**
Stanford University
ziyi.yang@stanford.edu

**Chenguang Zhu**
Microsoft Research
chezhu@microsoft.com

## Abstract

Due to widespread interest in machine translation and transfer learning, there are numerous algorithms for mapping multiple embeddings to a shared representation space. Recently, these algorithms have been studied in the setting of bilingual lexicon induction where one seeks to align the embeddings of a source and a target language such that translated word pairs lie close to one another in a common representation space. In this paper, we propose a method, Filtered Inner Product Projection (FIPP), for mapping embeddings to a common representation space. As semantic shifts are pervasive across languages and domains, FIPP first identifies the common geometric structure in both embeddings and then, only on the common structure, aligns the Gram matrices of these embeddings. FIPP aligns embeddings to isomorphic vector spaces even when the source and target embeddings are of differing dimensionalities. Additionally, FIPP provides computational benefits in ease of implementation and is faster to compute than current approaches. Following the baselines in Glavaš et al. (2019), we evaluate FIPP in the context of bilingual lexicon induction and downstream language tasks. We show that FIPP outperforms existing methods on the XLING (5K) BLI dataset and the XLING (1K) BLI dataset, when using a self-learning approach, while also providing robust performance across downstream tasks.

## 1 Introduction

The problem of aligning sets of embeddings, or high dimensional real valued vectors, is of great interest in natural language processing, with applications in machine translation and transfer learning, and shares connections to graph matching and assignment problems (Grave et al., 2019; Gold & Rangarajan, 1996). Aligning embeddings trained on corpora from different languages has led to improved performance of supervised and unsupervised word and sentence translation (Zou et al., 2013), sequence labeling (Zhang et al., 2016; Mayhew et al., 2017), and information retrieval (Vulić & Moens, 2015). Additionally, linguistic patterns have been studied using embedding alignment algorithms (Schlechtweg et al., 2019; Lauscher & Glavaš, 2019). Embedding alignments have also been shown to improve the performance of multilingual contextual representation models (i.e. mBERT), when used during intialization, on certain tasks such as multilingual document classification (Artetxe et al., 2020) Recently, algorithms using embedding alignments on the input token representations of contextual embedding models have been shown to provide efficient domain adaptation (Poerner et al., 2020). Lastly, aligned source and target input embeddings have been shown to improve the transferability of models learned on a source domain to a target domain (Artetxe et al., 2018a; Wang et al., 2018; Mogadala & Rettinger, 2016).

In the bilingual lexicon induction task, one seeks to learn a transformation on the embeddings of a source and a target language so that translated word pairs lie close to one another in the shared representation space. Specifically, one is given a small seed dictionary $D$ containing $c$ pairs of translated words, and embeddings for these word pairs in a source and a target language, $X_s \in \mathbb{R}^{c \times d}$

and $X_t \in \mathbb{R}^{c \times d}$. Using this seed dictionary, a transformation is learned on $X_s$ and $X_t$ with the objective that unseen translation pairs can be induced, often through nearest neighbors search.

Previous literature on this topic has focused on aligning embeddings by minimizing matrix or distributional distances (Grave et al., 2019; Jawanpuria et al., 2019; Joulin et al., 2018a). For instance, Mikolov et al. (2013a) proposed using Stochastic Gradient Descent (SGD) to learn a mapping, $\Omega$, to minimize the sum of squared distances between pairs of words in the seed dictionary $\|X_s^D \Omega - X_t^D\|_F^2$, which achieves high word translation accuracy for similar languages. Smith et al. (2017) and Artetxe et al. (2017) independently showed that a mapping with an additional orthogonality constraint, to preserve the geometry of the original spaces, can be solved with the closed form solution to the Orthogonal Procrustes problem, $\Omega^* = \arg\min_{\Omega \in O(d)} \|X_s^D \Omega - X_t^D\|_F$ where $O(d)$ denotes the group of $d$ dimensional orthogonal matrices. However, these methods usually require the dimensions of the source and target language embeddings to be the same, which often may not hold. Furthermore, due to semantic shifts across languages, it's often the case that a word and its translation may not co-occur with the same sets of words (Gulordava & Baroni, 2011). Therefore, seeking an alignment which minimizes all pairwise distances among translated pairs results in using information not common to both the source and target embeddings.

To address these problems, we propose Filtered Inner Product Projection (FIPP) for mapping embeddings from different languages to a shared representation space. FIPP aligns a source embedding $\mathfrak{X}_s \in \mathbb{R}^{n \times d_1}$ to a target embedding $\mathfrak{X}_t \in \mathbb{R}^{m \times d_2}$ and maps vectors in $\mathfrak{X}_s$ to the $\mathbb{R}^{d_2}$ space of $\mathfrak{X}_t$. Instead of word-level information, FIPP focuses on pairwise distance information, specified by the Gram matrices $X_s X_s^T$ and $X_t X_t^T$, where the rows of $X_s$ and $X_t$ correspond to embeddings for the $c$ pairs of source and target words from the seed dictionary. During alignment, FIPP tries to achieve the following two goals. First, it is desired that the aligned source embedding $\text{FIPP}(X_s) = \tilde{X}_s \in \mathbb{R}^{c \times d_2}$ be structurally close to the original source embedding to ensure that semantic information is retained and prevent against overfitting on the seed dictionary. This goal is reflected in the minimization of the *reconstruction loss*: $\|\tilde{X}_s \tilde{X}_s^T - X_s X_s^T\|_F^2$.

Second, as the usage of words and their translations vary across languages, instead of requiring $\tilde{X}_s$ to use all of the distance information from $X_t$, FIPP selects a filtered set $K$ of word pairs that have similar distances in both the source and target languages: $K = \{(i,j) \in D : |X_s X_s^T - X_t X_t^T|_{ij} \le \epsilon\}$. FIPP then minimizes a *transfer loss* on this set $K$, the squared difference in distances between the aligned source embeddings and the target embeddings: $\sum_{(i,j) \in K} (\tilde{X}_s[i]\tilde{X}_s[j]^T - X_t[i]X_t[j]^T)^2$.

We show FIPP can be efficiently solved using either low-rank semidefinite approximations or with stochastic gradient descent. Also, we formulate a least squares projection to infer aligned representations for words outside the seed dictionary and present a weighted Procrustes objective which recovers an orthogonal operator that takes into consideration the degree of structural similarity among translation pairs. The method is illustrated in Figure 1.

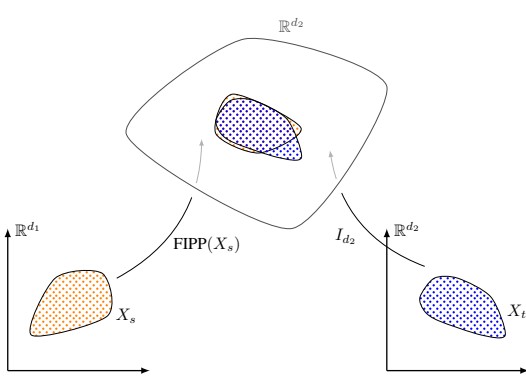

Figure 1: FIPP alignment of source and target embeddings, $X_s$ and $X_t$, to a common representation space. Note $X_s$ is modified using information from $X_t$ and mapped to $\mathbb{R}^{d_2}$ while $X_t$ is unchanged.

Compared to previous approaches, FIPP has improved generality, stability, and efficiency. First, since FIPP's alignment between the source and target embeddings is performed on Gram matrices, i.e. $X_s X_s^T$ and $X_t X_t^T \in \mathbb{R}^{c \times c}$, embeddings are not required to be of the same dimension and are projected to isomorphic vector spaces. This is particularly helpful for aligning embeddings trained on smaller corpora, such as in low resource domains, or compute-intensive settings where embeddings may have been compressed to lower dimensions. Secondly, alignment modifications made on filtered Gram matrices can incorporate varying constraints on alignment at the most granular level, pairwise distances. Lastly, FIPP is easy to implement as it involves only matrix

operations, is deterministic, and takes an order of magnitude less time to compute than either the best supervised (Joulin et al., 2018b) or unsupervised approach (Artetxe et al., 2018c) compared against.

We conduct a thorough evaluation of FIPP using baselines outlined in Glavaš et al. (2019) including bilingual lexicon induction with 5K and 1K supervision sets and downstream evaluation on MNLI Natural Language Inference and Ted CLDC Document Classification tasks. The rest of this paper is organized as follows. We discuss related work in Section 2. We introduce our FIPP model in Section 3 and usage for inference in Section 4. We present experimental results in Section 5 and further discuss findings in Section 6. We conclude the paper in Section 7.

## 2  RELATED WORK

### 2.1  DISTRIBUTIONAL METHODS FOR QUANTIFYING SEMANTIC SHIFTS

Prior work has shown that monolingual text corpora from different communities or time periods exhibit variations in semantics and syntax (Hamilton et al., 2016a;b). Word embeddings (Mikolov et al., 2013b; Pennington et al., 2014; Bojanowski et al., 2017) map words to representations in a continuous space with the objective that the inner product between any two words representations is approximately proportional to their probability of co-occurrence. By comparing pairwise distances in monolingual embeddings trained on separate corpora, one can quantify semantic shifts associated with biases, cultural norms, and temporal differences (Gulordava & Baroni, 2011; Sagi et al., 2011; Kim et al., 2014). Recently proposed metrics on embeddings compare all pairwise inner products of two embeddings, $E$ and $F$, of the form $\|EE^T - FF^T\|_F$ (Yin et al., 2018). While these metrics have been applied in quantifying monolingual semantic variation, they have not been explored in context of mapping embeddings to a common representation space or in multilingual settings.

### 2.2  CROSSLINGUAL EMBEDDING ALIGNMENT

The first work on this topic is by Mikolov et al. (2013a) who proposed using Stochastic Gradient Descent (SGD) to learn a mapping, $\Omega$, to minimize the sum of squared distances between pairs of words in the seed dictionary $\|X_s\Omega - X_t\|_F^2$, which achieves high word translation accuracy for similar languages. Smith et al. (2017) and Artetxe et al. (2017) independently showed that a mapping with an additional orthogonality constraint, to preserve the geometry of the original spaces, can be solved with the closed form solution to the Orthogonal Procrustes problem, $\Omega^* = \arg\min_{\Omega \in O(d)} \|X_s\Omega - X_t\|_F$. Dinu & Baroni (2015) worked on corrections to the "hubness" problem in embedding alignment, where certain word vectors may be close to many other word vectors, arising due to nonuniform density of vectors in the $R^d$ space. Smith et al. (2017) proposed the inverted softmax metric for inducing matchings between words in embeddings of different languages. Artetxe et al. (2016) studied the impact of normalization, centering and orthogonality constraints in linear alignment functions. Jawanpuria et al. (2019) presented a composition of orthogonal operators and a Mahalanobis metric of the form $UBV^T, U, V^T \in O(d), B \succ 0$ to account for observed correlations and moment differences between dimensions (Søgaard et al., 2018). Joulin et al. (2018a) proposed an alignment based on neighborhood information to account for differences in density and shape of embeddings in their respective $\mathbb{R}^d$ spaces. Artetxe et al. (2018c) outlined a framework which unifies many existing alignment approaches as compositions of matrix operations such as Orthogonal mappings, Whitening, and Dimensionality Reduction. Nakashole & Flauger (2018) found that locally linear maps vary between different neighborhoods in bilingual embedding spaces which suggests that nonlinearity is beneficial in global alignments. Nakashole (2018) proposed an alignment method using neighborhood sensitive maps which shows strong performance on dissimilar language pairs. Patra et al. (2019) proposed a novel hub filtering method and a semi-supervised alignment approach based on distributional matching. Mohiuddin et al. (2020) learned a non-linear mapping in the latent space of two independently pre-trained autoencoders which provide strong performance on well-studied BLI tasks. A recent method, most similar to ours, Glavaš & Vulić (2020) utilizes non-linear mappings to find a translation vector for each source and target embedding using the cosine similarity and euclidean distances between nearest neighbors and corresponding translations. In the unsupervised setting, where a bilingual seed dictionary is not provided, approaches using adversarial learning, distributional matching, and noisy self-supervision have been used to concurrently learn a matching and an alignment between embeddings (Cao et al., 2016; Zhang et al., 2017; Hoshen

& Wolf, 2018; Grave et al., 2019; Artetxe et al., 2017; 2018b; Alvarez-Melis & Jaakkola, 2018). Discussion on unsupervised approaches is included in Appendix Section I.

# 3 FILTERED INNER PRODUCT PROJECTION (FIPP)

## 3.1 FILTERED INNER PRODUCT PROJECTION OBJECTIVE

In this section, we introduce Filtered Inner Product Projection (FIPP), a method for aligning embeddings in a shared representation space. FIPP aligns a source embedding $\mathfrak{X}_s \in \mathbb{R}^{n \times d_1}$ to a target embedding $\mathfrak{X}_t \in \mathbb{R}^{m \times d_2}$ and projects vectors in $\mathfrak{X}_s$ to $\tilde{\mathfrak{X}}_s \in \mathbb{R}^{n \times d_2}$. Let $X_s \in \mathbb{R}^{c \times d_1}$ and $X_t \in \mathbb{R}^{c \times d_2}$ be the source and target embeddings for pairs in the seed dictionary $D$, $|D| = c \ll min(n, m)$. FIPP's objective is to minimize a linear combination of a reconstruction loss, which regularizes changes in the pairwise inner products of the source embedding, and a transfer loss, which aligns the source and target embeddings on common portions of their geometries.

$$\min_{\tilde{X}_s \in \mathbb{R}^{c \times d_2}} \overbrace{\|\tilde{X}_s \tilde{X}_s^T - X_s X_s^T\|_F^2}^{\text{Reconstruction Loss}} + \lambda \underbrace{\|\Delta^\epsilon \circ (\tilde{X}_s \tilde{X}_s^T - X_t X_t^T)\|_F^2}_{\text{Transfer Loss}} \tag{1}$$

where $\lambda, \epsilon \in \mathbb{R}^+$ are tunable scalar hyperparameters whose effects are discussed in Section E, $\circ$ is the Hadamard product, and $\Delta^\epsilon$ is a binary matrix discussed in 3.1.2.

### 3.1.1 RECONSTRUCTION LOSS

Due to the limited, noisy supervision in our problem setting, an alignment should be regularized against overfitting. Specifically, the aligned space needs to retain a similar geometric structure to the original source embeddings; this has been enforced in previous works by ensuring that alignments are close to orthogonal mappings (Mikolov et al., 2013a; Joulin et al., 2018a; Jawanpuria et al., 2019). As $\tilde{X}_s$ and $X_s$ can be of differing dimensionality, we check structural similarity by comparing pairwise inner products, captured by a reconstruction loss known as the PIP distance or Global Anchor Metric: $\|\tilde{X}_s \tilde{X}_s^T - X_s X_s^T\|_F^2$ (Yin & Shen, 2018; Yin et al., 2018).

**Theorem 1.** *Suppose $E \in \mathbb{R}^{n \times d}$, $F \in \mathbb{R}^{n \times d}$ are two matrices with orthonormal columns and $\Omega^* = \arg\min_{\Omega \in O(d)} \|E\Omega - F\|_F$. It follows that (Yin et al., 2018):*

$$\|E\Omega^* - F\| \leq \|EE^T - FF^T\| \leq \sqrt{2}\|E\Omega^* - F\|. \tag{2}$$

This metric has been used in quantifying semantic shifts and has been shown Yin et al. (2018) to be equivalent to the residual of the Orthogonal Procrustes problem up to a small constant factor, as seen in Theorem 1. Note that the PIP distance is invariant to orthogonal operations such as rotations which are known to be present in unaligned embeddings.

### 3.1.2 TRANSFER LOSS

In aligning $X_s$ to $X_t$, we should seek to only utilize common geometric information between the two embedding spaces. We propose a simple approach, although FIPP can admit other forms of filtering mechanisms, denoted as inner product filtering where we only utilize pairwise distances similar in both embedding spaces as defined by a threshold $\epsilon$. Specifically, compute a matrix $\Delta^\epsilon \in \{0,1\}^{c \times c}$ where $\Delta_{ij}^\epsilon$ is an indicator on whether $|X_{s,i}X_{s,j}^T - X_{t,i}X_{t,j}^T| < \epsilon$. In this form, $\epsilon$ is a hyperparameter which determines how close pairwise distances must be in the source and target embeddings to be deemed similar. We then define a transfer loss as being the squared difference between the converted source embedding $\tilde{X}_s$ and target embedding $X_t$, but only on pairs of words in $K$: $\|\Delta^\epsilon \circ (\tilde{X}_s \tilde{X}_s^T - X_t X_t^T)\|_F^2$, where $\circ$ is the Hadamard product. The FIPP objective is a linear combination of the reconstruction and transfer losses.

## 3.2 APPROXIMATE SOLUTIONS TO THE FIPP OBJECTIVE

### 3.2.1 SOLUTIONS USING LOW-RANK SEMIDEFINITE APPROXIMATIONS

Denote the Gram matrices $G^s \triangleq X_s X_s^T$, $G^t \triangleq X_t X_t^T$ and $\tilde{G}^s \triangleq \tilde{X}_s \tilde{X}_s^T$.

**Lemma 2.** *The matrix $G^*$ which minimizes the FIPP objective for a fixed $\lambda$ and $\epsilon$ has entries:*

$$G_{ij}^* = \begin{cases} \frac{(X_s X_s^T)_{ij} + \lambda (X_t X_t^T)_{ij}}{1+\lambda}, & \text{if } (i,j) \in K \\ (X_s X_s^T)_{ij}, & \text{otherwise} \end{cases} \tag{3}$$

*Proof.* For a fixed $\lambda$ and $\epsilon$, $\mathcal{L}_{FIPP,\lambda,\epsilon}(\tilde{X}_s \tilde{X}_s^T)$ can be decomposed as follows:

$$\begin{aligned} \mathcal{L}_{FIPP,\lambda,\epsilon}(\tilde{X}_s \tilde{X}_s^T) = &\|\tilde{X}_s \tilde{X}_s^T - X_s X_s^T\|_F^2 + \lambda \|\Delta^\epsilon \circ (\tilde{X}_s \tilde{X}_s^T - X_t X_t^T)\|_F^2 \\ = &\sum_{i,j \in K} ((\tilde{G}_{ij}^s - G_{ij}^s)^2 + \lambda(\tilde{G}_{ij}^s - G_{ij}^t)^2) + \sum_{i,j \notin K} (\tilde{G}_{ij}^s - G_{ij}^s)^2 \end{aligned} \tag{4}$$

By taking derivatives with respect to $\tilde{G}_{ij}^s$, the matrix $G^*$ which minimizes $\mathcal{L}_{FIPP,\lambda,\epsilon}(\cdot)$ is:

$$G^* = \underset{\tilde{X}_s \tilde{X}_s^T \in \mathbb{R}^{c \times c}}{\arg\min} \mathcal{L}_{FIPP,\lambda,\epsilon}(\tilde{X}_s \tilde{X}_s^T), \quad G_{ij}^* = \begin{cases} \frac{(X_s X_s^T)_{ij} + \lambda (X_t X_t^T)_{ij}}{1+\lambda}, & \text{if } (i,j) \in K \\ (X_s X_s^T)_{ij}, & \text{otherwise} \end{cases} \tag{5}$$

$\square$

We now have the matrix $G^* \in \mathbb{R}^{c \times c}$ which minimizes the FIPP objective. However, for $G^*$ to be a valid Gram matrix, it is required that $G^* \in S_+^{c \times c}$, the set of symmetric Positive Semidefinite matrices. Additionally, to recover an $\tilde{X}_s \in \mathbb{R}^{c \times d_2}$ such that $\tilde{X}_s \tilde{X}_s^T = G^*$, we must have $Rank(G^*) \le d_2$.

Note that $G^*$ is symmetric by construction since the set $K$ is commutative and $G^s, G^t$ are symmetric. However, $G^*$ is not necessarily positive semidefinite nor is it necessarily true that $Rank(G^*) \le d_2$. Therefore, to recover an aligned embedding $\tilde{X}_s \in \mathbb{R}^{c \times d_2}$, we perform a rank-constrained semidefinite approximation to find $\min_{\tilde{X}_s \in \mathbb{R}^{c \times d_2}} \|\tilde{X}_s \tilde{X}_s^T - G^*\|_F$.

**Theorem 3.** *Let $G^* = Q\Lambda Q^T$ be the Eigendecomposition of $G^*$. A matrix $\tilde{X}_s \in \mathbb{R}^{m \times d_2}$ which minimizes $\|\tilde{X}_s \tilde{X}_s^T - G^*\|_F$ is given by $\sum_{i=1, \lambda_i \ge 0}^{d_2} \lambda_i^{\frac{1}{2}} q_i$, where $\lambda_i$ and $q_i$ are the $i^{th}$ largest eigenvalue and corresponding eigenvector.*

*Proof.* Since $G^* \in S^{c \times c}$, its Eigendecomposition is $G^* = Q\Lambda Q^T$ where $Q$ is orthonormal. Let $\tilde{\lambda}, \tilde{q}$ be the $d_2$ largest nonnegative eigenvalues in $\Lambda$ and their corresponding eigenvectors; additionally, denote the complementary eigenvalues and associated eigenvectors as $\tilde{\lambda}^\perp = \Lambda \setminus \tilde{\lambda}, \tilde{q}^\perp = Q \setminus \tilde{q}$. Using the Eckart–Young–Mirsky Theorem for the Frobenius norm (Kishore Kumar & Schneider, 2017), note that for $G \in S_+^{c \times c}, Rank(G) \le d_2$; $\|G^* - G\|_F \ge \|\tilde{q}^\perp \tilde{\lambda}^\perp \tilde{q}^{\perp T}\|_F = \sum_{\lambda_i \in \tilde{\lambda}^\perp} \lambda_i^{\frac{1}{2}}$ and that $\|G^* - G\|_F$ is minimized for $G = \tilde{q} \tilde{\lambda} \tilde{q}^T$. Using this result, we can recover $\tilde{X}_s$:

$$\underset{\substack{G \in S_+^{c \times c}, \\ Rank(G) \le d_2}}{\arg\min} \|G^* - G\|_F = \sum_{\lambda_i \in \tilde{\lambda}} (\lambda_i^{\frac{1}{2}} q_i)(\lambda_i^{\frac{1}{2}} q_i)^T = \tilde{X}_s \tilde{X}_s^T \tag{6}$$

Using the above matrix approximation, we find our aligned embedding $\tilde{X}_s = \sum_{\lambda_i \in \tilde{\lambda}} \lambda_i^{\frac{1}{2}} q_i$, a minimizer of $\|\tilde{X}_s \tilde{X}_s^T - G^*\|_F$. $\square$

Due to the rank constraint on $G$, we are only interested in the $d_2$ largest eigenvalues and corresponding eigenvectors which incurs a complexity of $\mathcal{O}(d_2 c^2)$ using power iteration (Panju, 2011).

### 3.2.2 SOLUTIONS USING STOCHASTIC GRADIENT DESCENT

Alternatively, solutions to the FIPP objective can be obtained using Stochastic Gradient Descent (SGD). This requires defining a single variable $\tilde{X}_s \in \mathbb{R}^{c \times d_2}$ over which to optimize. We find that the solutions obtained after convergence of SGD are close, with respect to the Frobenius norm, to those obtained with low rank PSD approximations up to a rotation. However, the complexity of solving FIPP using SGD is $\mathcal{O}(Tc^2)$, where $T$ is the number of training epochs. Empirically we find $T > d_2$ for SGD convergence and, as a result, this approach incurs a complexity greater than that of low-rank semidefinite approximations.

### 3.3 ISOTROPIC PREPROCESSING

Common preprocessing steps used by previous approaches (Joulin et al., 2018a; Artetxe et al., 2018a), involve normalizing the rows of $\mathfrak{X}_s, \mathfrak{X}_t$ to have an $\ell_2$ norm of 1 and demeaning columns. The transfer loss of the FIPP objective makes direct comparisons on the Gram matrices, $X_s X_s^T$ and $X_t X_t^T$, of the source and target embeddings for words in the seed dictionary. To reduce the influence of dimensional biases between $X_s$ and $X_t$ and ensure words are weighted equally during alignment, it is desired that $X_s$ and $X_t$ be isotropic - i.e. $Z(c) = \sum_{i \in X_s} \exp c^T X_s[i]$ is approximately constant for any unit vector $c$ (Arora et al., 2016). Mu & Viswanath (2018) find that a first order approximation to enforce isotropic behavior is achieved by column demeaning while a second order approximation is obtained by the removal of the top PCA components. In FIPP, we apply this simple pre-processing approach by removing the top PCA component of $X_s$ and $X_t$. Empirically, the distributions of inner products between a source and target embedding can differ substantially when not preprocessed, rendering a substandard alignment, which is discussed further in the Appendix Section G.

## 4 INFERENCE AND ALIGNMENT

### 4.1 INFERENCE WITH LEAST SQUARES PROJECTION

To infer aligned source embeddings for words outside of the supervision dictionary, we make the assumption that source words not used for alignment should preserve their distances to those in the seed dictionary in their respective spaces, i.e., $X_s \mathfrak{X}_s^T \approx \tilde{X}_s \tilde{\mathfrak{X}}_s^T$. Using this assumption, we formulate a least squares projection (Boyd & Vandenberghe, 2004) on an overdetermined system of equations to recover $\tilde{\mathfrak{X}}_s^T$: $\tilde{\mathfrak{X}}_s^T = (\tilde{X}_s^T \tilde{X}_s)^{-1} \tilde{X}_s^T X_s \mathfrak{X}_s^T$.

### 4.2 WEIGHTED ORTHOGONAL PROCRUSTES

As $\tilde{X}_s \in \mathbb{R}^{c \times d_2}$ has been optimized only with concern for its inner products, $\tilde{X}_s$ must be rotated so that it's basis matches that of $X_t$. We propose a weighted variant of the Orthogonal Procrustes solution to account for differing levels of translation uncertainty among pairs in the seed dictionary, which may arise due to polysemy, semantic shifts and translation errors. In Weighted Least Squares problems, an inverse variance-covariance weighting $W$ is used (Strutz, 2010; Brignell et al., 2015) to account for differing levels of measurement uncertainty among samples. We solve a weighted Procrustes objective, where measurement error is approximated as the transfer loss for each translation pair, $W_{ii}^{-1} = \|\tilde{X}_s[i]X_s^T - X_t[i]X_t^T\|^2$:

$$\text{SVD}((WX_t)^T W \tilde{X}_s) = U\Sigma V^T, \Omega^W = \underset{\Omega \in O(d_2)}{\arg\min} \|W(\tilde{X}_s \Omega - X_t)\|_F^2 = UV^T, \tag{7}$$

where $O(d_2)$ is the group of $d_2 \times d_2$ orthogonal matrices. The rotation $\Omega^W$ is then applied to $\tilde{X}_s$.

## 5 EXPERIMENTATION

In this section, we report bilingual lexicon induction results from the XLING dataset and downstream experiments performed on the MNLI Natural Language Inference and TED CLDC tasks.

### 5.1 XLING BILINGUAL LEXICON INDUCTION

The XLing BLI task dictionaries constructed by Glavaš et al. (2019) include all 28 pairs between 8 languages in different language families, Croatian (HR), English (EN), Finnish (FI), French (FR), German (DE), Italian (IT), Russian (RU), and Turkish (TR). The dictionaries use the same vocabulary across languages and are constructed based on word frequency, to reduce biases known to exist in other datasets (Kementchedjhieva et al., 2019). We evaluate FIPP across the all language pairs using a supervision dictionary of size 5K and 1K. On 5K dictionaries, FIPP outperforms other approaches on 22 of 28 language pairs. On 1K dictionaries, FIPP outperforms other approaches on 23 of 28

language pairs when used along with a Self-Learning Framework (denoted as FIPP + SL) discussed in Appendix Section C. Our code for FIPP is open-source and available on Github [1].

### 5.1.1 Bilingual Lexicon Induction: XLING 1K

| Method | VecMap | ICP | CCA | PROC[‡] | PROC-B | DLV | RCSLS[‡] | FIPP | FIPP + SL |
|---|---|---|---|---|---|---|---|---|---|
| DE-FI | 0.302 | 0.251 | 0.241 | 0.264 | **0.354** | 0.259 | 0.288 | 0.296 | 0.345 |
| DE-FR | 0.505 | 0.454 | 0.422 | 0.428 | 0.511 | 0.384 | 0.459 | 0.463 | **0.530** |
| EN-DE | 0.521 | 0.486 | 0.458 | 0.458 | 0.521 | 0.454 | 0.501 | 0.513 | **0.568** |
| EN-HR | 0.268 | 0.000 | 0.218 | 0.225 | 0.296 | 0.225 | 0.267 | 0.275 | **0.320** |
| FI-HR | 0.280 | 0.208 | 0.167 | 0.187 | 0.263 | 0.184 | 0.214 | 0.243 | **0.304** |
| FI-IT | 0.355 | 0.263 | 0.232 | 0.247 | 0.328 | 0.244 | 0.272 | 0.309 | **0.372** |
| HR-IT | **0.389** | 0.045 | 0.240 | 0.247 | 0.343 | 0.245 | 0.275 | 0.318 | **0.389** |
| IT-FR | 0.667 | 0.629 | 0.612 | 0.615 | 0.665 | 0.585 | 0.637 | 0.639 | **0.678** |
| RU-IT | 0.463 | 0.394 | 0.352 | 0.360 | 0.466 | 0.358 | 0.383 | 0.413 | **0.489** |
| TR-RU | 0.200 | 0.119 | 0.146 | 0.168 | 0.230 | 0.161 | 0.191 | 0.205 | **0.248** |
| Avg. (All 28 Lang. Pairs) | **0.375** | 0.253 | 0.289 | 0.299 | 0.379 | 0.289 | 0.331 | 0.344 | **0.406** |

Table 1: Mean Average Precision (MAP) of alignment methods on a subset of XLING BLI with 1K supervision dictionaries, retrieval method is nearest neighbors. Benchmark results obtained from Glavaš et al. (2019) in which (‡) PROC was evaluated without preprocessing and RCSLS was evaluated with Centering + Length Normalization (C+L) preprocessing. Full results for XLING 1K can be found in Appendix Table 8.

### 5.1.2 Bilingual Lexicon Induction: XLING 5K

| Method | VecMap | MUSE | ICP | CCA | PROC[‡] | PROC-B* | DLV | RCSLS[‡] | FIPP |
|---|---|---|---|---|---|---|---|---|---|
| DE-FI | 0.302 | 0.000 | 0.251 | 0.353 | 0.359 | 0.362 | 0.357 | **0.395** | 0.389 |
| DE-FR | 0.505 | 0.005 | 0.454 | 0.509 | 0.511 | 0.514 | 0.506 | 0.536 | **0.543** |
| EN-DE | 0.521 | 0.520 | 0.486 | 0.542 | 0.544 | 0.532 | 0.545 | 0.580 | **0.590** |
| EN-HR | 0.268 | 0.000 | 0.000 | 0.325 | 0.336 | 0.336 | 0.334 | 0.375 | **0.382** |
| FI-HR | 0.280 | 0.228 | 0.208 | 0.288 | 0.294 | 0.293 | 0.294 | 0.321 | **0.335** |
| FI-IT | 0.355 | 0.000 | 0.263 | 0.353 | 0.355 | 0.348 | 0.356 | 0.388 | **0.407** |
| HR-IT | 0.389 | 0.000 | 0.045 | 0.366 | 0.364 | 0.368 | 0.366 | 0.399 | **0.415** |
| IT-FR | 0.667 | 0.662 | 0.629 | 0.668 | 0.669 | 0.664 | 0.665 | 0.682 | **0.684** |
| RU-IT | 0.463 | 0.450 | 0.394 | 0.474 | 0.474 | 0.476 | 0.475 | 0.491 | **0.503** |
| TR-RU | 0.200 | 0.000 | 0.119 | 0.285 | 0.290 | 0.262 | 0.289 | **0.324** | 0.319 |
| Avg. (All 28 Lang. Pairs) | **0.375** | 0.183 | 0.253 | 0.400 | 0.405 | 0.398 | 0.403 | 0.437 | **0.442** |

Table 2: Mean Average Precision (MAP) of alignment methods on a subset of XLING BLI with 5K supervision dictionaries, retrieval method is nearest neighbors. Benchmark results obtained from Glavaš et al. (2019) in which (*) Proc-B was reported using a 3K seed dictionary, (‡) PROC was evaluated without preprocessing and RCSLS was evaluated with (C+L) preprocessing. Full results for XLING 5K can be found in Appendix Table 7.

## 5.2 Downstream Evaluations

### 5.2.1 TED CLDC - Document Classification

The TED-CLDC corpus (Hermann & Blunsom, 2014) contains cross-lingual documents across 15 topics and 12 language pairs. Following the evaluation of Glavaš et al. (2019), a simple CNN based classifier is trained and evaluated over each topic for language pairs included in our BLI evaluations (EN-DE, EN-FR, EN-IT, EN-RU, and EN-TR). RCSLS outperforms other methods overall and on DE, FR, and RU while FIPP performs best on IT and TR.

---

[1] https://github.com/vinsachi/FIPPCLE

| Method | VecMap | MUSE | ICP | GWA | PROC | PROC-B | DLV | RCSLS | FIPP |
|--------|--------|------|-----|-----|------|--------|-----|-------|------|
| DE | 0.433 | 0.288 | 0.492 | 0.180* | 0.345 | 0.352 | 0.299 | **0.588** | 0.520 |
| FR | 0.316 | 0.223 | 0.254 | 0.209* | 0.239 | 0.210 | 0.175 | **0.540** | 0.433 |
| IT | 0.333 | 0.198* | 0.457* | 0.206* | 0.310 | 0.218 | 0.234 | 0.451 | **0.459** |
| RU | 0.504 | 0.226* | 0.362 | 0.151* | 0.251 | 0.186 | 0.375 | **0.527** | 0.481 |
| TR | 0.439 | 0.264* | 0.175 | 0.173* | 0.190 | 0.310 | 0.208 | 0.447 | **0.491** |
| Avg. | 0.405 | 0.240 | 0.348 | 0.184 | 0.267 | 0.255 | 0.258 | **0.510** | 0.477 |

Table 3: TED-CLDC micro-averaged $F_1$ scores using a CNN model with embeddings from different alignment methods. Evaluation follows Glavaš et al. (2019), (*) signifies language pairs for which unsupervised methods were unable to yield successful runs.

### 5.2.2 MNLI - Natural Language Inference

The multilingual XNLI corpus introduced by Conneau et al. (2018), based off the English only MultiNLI (Williams et al., 2018), includes 5 of the 8 languages used in BLI: EN, DE, FR, RU, and TR. We perform the same evaluation as Glavaš et al. (2019) by training an ESIM model (Chen et al., 2017) using EN word embeddings from a shared EN-L2 embedding space for L2 $\in$ {DE, FR, RU, TR}. The trained model is then evaluated without further training on the L2 XNLI test set using L2 embeddings from the shared space. The bootstrap Procrustes approach (Glavaš et al., 2019) outperforms other methods narrowly while RCSLS performs worst despite having high BLI accuracy.

| Method | VecMap | MUSE | ICP | GWA | PROC | PROC-B | DLV | RCSLS | FIPP |
|--------|--------|------|-----|-----|------|--------|-----|-------|------|
| EN-DE | 0.604 | 0.611 | 0.580 | 0.427* | 0.607 | **0.615** | 0.614 | 0.390 | 0.603 |
| EN-FR | **0.613** | 0.536 | 0.510 | 0.383* | 0.534 | 0.532 | 0.556 | 0.363 | 0.509 |
| EN-RU | 0.574 | 0.363* | 0.572 | 0.376* | 0.585 | **0.599** | 0.571 | 0.399 | 0.577 |
| EN-TR | 0.534 | 0.359* | 0.400* | 0.359* | 0.568 | 0.573 | **0.579** | 0.387 | 0.547 |
| Avg. | **0.581** | 0.467 | 0.516 | 0.386 | 0.574 | **0.580** | 0.571 | 0.385 | 0.559 |

Table 4: MNLI test accuracy using an ESIM model with embeddings from different alignment methods. Evaluation follows Glavaš et al. (2019), (*) signifies language pairs for which unsupervised methods were unable to yield successful runs.

## 6 Discussion

### 6.1 Runtime Comparison

| Method | FIPP | FIPP+SL | VecMap | RCSLS | Proc-B |
|--------|------|---------|--------|-------|--------|
| CPU | 23s | - | 15,896s | 387s | 885s |
| GPU | - | 176s | 612s | - | - |

Table 5: Average alignment time; sup. approaches use a 5K dictionary. FIPP+SL augments with additional 10K samples.

An advantage of FIPP compared to existing methods is it's computational efficiency. We provide a runtime comparison of FIPP and the best performing unsupervised (Artetxe et al., 2018c) and supervised (Joulin et al., 2018b) alignment methods on the XLING 5K BLI tasks along with Proc-B (Glavaš et al., 2019) a supervised method which performs best out of the compared approaches on 1K seed dictionaries. The average execution time of alignment on 3 runs of the 'EN-DE' XLING 5K dictionary is provided in Table 4. The implementation used for RCSLS is from Facebook's fastText repo [2] with default parameters and the VecMap implementation is from the author's repo [3] with and without the 'cuda' flag. Proc-B is implemented from the XLING-Eval repo [4] with default parameters. Hardware specifications are 2 Nvidia GeForce GTX 1080 Ti GPUs, 12 Intel Core i7-6800K processors, and 112GB RAM.

---

[2] https://github.com/facebookresearch/fastText/tree/master/alignment

[3] https://github.com/artetxem/vecmap

[4] https://github.com/codogogo/xling-eval

## 6.2 ALIGNMENT OF EMBEDDINGS OF DIFFERING DIMENSIONALITY

### 6.2.1 COMPARISON OF GRAM MATRIX SPECTRUM

In this section, we compute alignments with previous methods, a modification of Procrustes and RCSLS (Joulin et al., 2018b), and FIPP on an English Embedding $\in \mathbb{R}^{200}$ to a German Embedding $\in \mathbb{R}^{300}$. In Figure 2, we plot the spectrum of the Gram Matrices for the aligned embeddings from each method and the target German Embedding. While the FIPP aligned embedding is isomorphic to the target German Embedding $\in \mathbb{R}^{300}$, other methods produce a rank deficient aligned embedding whose spectrum deviates from the target embedding. The null space of aligned source embeddings, for methods other than FIPP, is of dimension $d_2 - d_1$. We note that issues can arise when learning and transferring models on embeddings from different rank vector spaces. For regularized models transferred from the source to the target space, at least $d_2 - d_1$ column features of the target embedding will not be utilized. Meanwhile, models

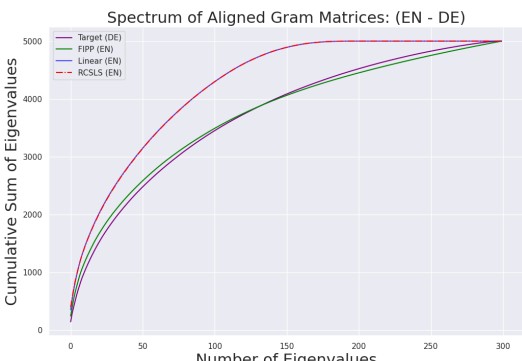

Figure 2: Spectrum of Aligned EN Embeddings $\in \mathbb{R}^{200}$ and DE Embeddings $\in \mathbb{R}^{300}$. Spectrums of Original, RCSLS and Linear EN embeddings are all approx. equivalent.

transferred from the target space to the source space will exhibit bias associated with model parameters corresponding to the source embedding's null space.

### 6.2.2 BLI PERFORMANCE

| $d_1$ | Linear | CCA | FIPP |
|-----|--------|-------|-------|
| 300 | 0.544 | 0.542 | 0.590 |
| 250 | 0.524 | 0.529 | 0.574 |
| 200 | 0.486 | 0.501 | 0.543 |

Table 6: XLING 5K (EN-DE) MAP for Linear, RCSLS, and FIPP alignment methods on embeddings of differing dimensionality.

FIPP is able to align embeddings of different dimensionalities to isomorphic vector spaces unlike competing approaches (Joulin et al., 2018b; Artetxe et al., 2018c). We evaluate BLI performance on embeddings of different dimensionalities for the EN-DE, (English, German), language pair. We assume that $d_1 \leq d_2$ and align EN embeddings with dimensions $\in \{200, 250, 300\}$ to a DE embedding of dimension 300 and compare the performance of FIPP with CCA, implemented in scikit-learn[5] as an iterative estimation of partial least squares, and the best linear transform with orthonormal rows, $\Omega^* = \arg\min_{\Omega \in \mathbb{R}^{d_1 \times d_2}, \Omega\Omega^T = I_{d_1}} \|X_s\Omega - X_t\|_F$, equivalent to the Orthogonal Procrustes solution when $d_1 = d_2$. Both the Linear and FIPP methods map $X_s$ and $X_t$ to dimension $d_2$ while the CCA method maps both embeddings to $min(d_1, d_2)$ which may be undesirable. While the performance of all methods decreases as $d_2 - d_1$ increases, the relative performance gap between FIPP and other approaches is maintained.

## 7 CONCLUSION

In this paper, we introduced Filtered Inner Product Projection (FIPP), a method for aligning multiple embeddings to a common representation space using pairwise inner product information. FIPP accounts for semantic shifts and aligns embeddings only on common portions of their geometries. Unlike previous approaches, FIPP aligns embeddings to equivalent rank vector spaces regardless of their dimensions. We provide two methods for finding approximate solutions to the FIPP objective and show that it can be efficiently solved even in the case of large seed dictionaries. We evaluate FIPP on the task of bilingual lexicon induction using the XLING (5K) dataset and the XLING (1K) dataset, on which it achieves state-of-the-art performance on most language pairs. Our method provides a novel efficient approach to the problem of shared representation learning.

---

[5]https://scikit-learn.org/stable/modules/generated/sklearn.cross_decomposition.CCA.html

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

## A  FULL BLI EXPERIMENTATION - XLING (1K) AND XLING (5K)

In Table 6 below, we provide experimental results for FIPP using 1K seed dictionaries. Unlike in case of a 5K supervision set, FIPP is outperformed by the bootstrapped Procrustes method (Glavaš et al., 2019) and the unsupervised VecMap approach (Artetxe et al., 2018c). However, the addition of a self learning framework, detailed in Section C, to FIPP (FIPP + SL) results in performance greater than compared methods albeit at the cost of close to a 8x increase in computation time, from 23s to 176s, and the requirement of a GPU. Other well performing methods for XLING 1K, Proc-B (Glavaš et al., 2019) and VecMap (Artetxe et al., 2018c), also use self-learning frameworks; further analysis is required to understand the importance of self-learning frameworks in the case of small (or no) seed dictionary.

The methods compared against were originally proposed in: VecMap (Artetxe et al., 2018c), MUSE (Lample et al., 2018), ICP (Hoshen & Wolf, 2018), CCA (Faruqui & Dyer, 2014), GWA (Alvarez-Melis & Jaakkola, 2018), PROC (Mikolov et al., 2013a), PROC-B (Glavaš et al., 2019), DLV (Ruder et al., 2018), and RCSLS (Joulin et al., 2018b).

| Method | VecMap | ICP | CCA | PROC‡ | PROC-B | DLV | RCSLS‡ | FIPP | FIPP + SL |
|--------|--------|-----|-----|-------|--------|-----|--------|------|-----------|
| DE-FI | 0.302 | 0.251 | 0.241 | 0.264 | **0.354** | 0.259 | 0.288 | 0.296 | 0.345 |
| DE-FR | 0.505 | 0.454 | 0.422 | 0.428 | 0.511 | 0.384 | 0.459 | 0.463 | **0.530** |
| DE-HR | 0.300 | 0.240 | 0.206 | 0.225 | 0.306 | 0.222 | 0.262 | 0.268 | **0.312** |
| DE-IT | 0.493 | 0.447 | 0.414 | 0.421 | 0.507 | 0.420 | 0.453 | 0.482 | **0.526** |
| DE-RU | 0.322 | 0.245 | 0.308 | 0.323 | **0.392** | 0.325 | 0.361 | 0.359 | 0.368 |
| DE-TR | 0.253 | 0.215 | 0.153 | 0.169 | 0.250 | 0.167 | 0.201 | 0.215 | **0.275** |
| EN-DE | 0.521 | 0.486 | 0.458 | 0.458 | 0.521 | 0.454 | 0.501 | 0.513 | **0.568** |
| EN-FI | 0.292 | 0.262 | 0.259 | 0.271 | 0.360 | 0.271 | 0.306 | 0.314 | **0.397** |
| EN-FR | 0.626 | 0.613 | 0.582 | 0.579 | 0.633 | 0.546 | 0.612 | 0.601 | **0.666** |
| EN-HR | 0.268 | 0.000 | 0.218 | 0.225 | 0.296 | 0.225 | 0.267 | 0.275 | **0.320** |
| EN-IT | 0.600 | 0.577 | 0.538 | 0.535 | 0.605 | 0.537 | 0.565 | 0.591 | **0.638** |
| EN-RU | 0.323 | 0.259 | 0.336 | 0.352 | 0.419 | 0.353 | 0.401 | 0.399 | **0.439** |
| EN-TR | 0.288 | 0.000 | 0.218 | 0.225 | 0.301 | 0.221 | 0.275 | 0.292 | **0.360** |
| FI-FR | **0.368** | 0.000 | 0.230 | 0.239 | 0.329 | 0.209 | 0.269 | 0.274 | 0.366 |
| FI-HR | 0.280 | 0.208 | 0.167 | 0.187 | 0.263 | 0.184 | 0.214 | 0.243 | **0.304** |
| FI-IT | 0.355 | 0.263 | 0.232 | 0.247 | 0.328 | 0.244 | 0.272 | 0.309 | **0.372** |
| FI-RU | 0.312 | 0.231 | 0.214 | 0.233 | 0.315 | 0.225 | 0.257 | 0.285 | **0.346** |
| HR-FR | **0.402** | 0.282 | 0.238 | 0.248 | 0.335 | 0.214 | 0.281 | 0.283 | 0.380 |
| HR-IT | **0.389** | 0.045 | 0.240 | 0.247 | 0.343 | 0.245 | 0.275 | 0.318 | **0.389** |
| HR-RU | 0.376 | 0.309 | 0.256 | 0.269 | 0.348 | 0.264 | 0.291 | 0.318 | **0.380** |
| IT-FR | 0.667 | 0.629 | 0.612 | 0.615 | 0.665 | 0.585 | 0.637 | 0.639 | **0.678** |
| RU-FR | 0.463 | 0.000 | 0.344 | 0.352 | 0.467 | 0.320 | 0.381 | 0.383 | **0.486** |
| RU-IT | 0.463 | 0.394 | 0.352 | 0.360 | 0.466 | 0.358 | 0.383 | 0.413 | **0.489** |
| TR-FI | 0.246 | 0.173 | 0.151 | 0.169 | 0.247 | 0.161 | 0.194 | 0.200 | **0.280** |
| TR-FR | 0.341 | 0.000 | 0.213 | 0.215 | 0.305 | 0.194 | 0.247 | 0.251 | **0.342** |
| TR-HR | 0.223 | 0.138 | 0.134 | 0.148 | 0.210 | 0.144 | 0.170 | 0.184 | **0.241** |
| TR-IT | 0.332 | 0.243 | 0.202 | 0.211 | 0.298 | 0.209 | 0.246 | 0.263 | **0.335** |
| TR-RU | 0.200 | 0.119 | 0.146 | 0.168 | 0.230 | 0.161 | 0.191 | 0.205 | **0.248** |
| AVG | **0.375** | 0.253 | 0.289 | 0.299 | 0.379 | 0.289 | 0.331 | 0.344 | **0.406** |

Table 7: Mean Average Precision (MAP) of alignment methods on XLING with 1K supervision dictionaries, retrieval method is nearest neighbors. Benchmark results obtained from Glavaš et al. (2019) in which (‡) PROC was evaluated without preprocessing and RCSLS was evaluated with Centering + Length Normalization (C+L) preprocessing

| Method | VecMap | MUSE | ICP | CCA | PROC‡ | PROC-B* | DLV | RCSLS‡ | FIPP |
|--------|--------|------|-----|-----|-------|---------|-----|--------|------|
| DE-FI | 0.302 | 0.000 | 0.251 | 0.353 | 0.359 | 0.362 | 0.357 | **0.395** | 0.389 |
| DE-FR | 0.505 | 0.005 | 0.454 | 0.509 | 0.511 | 0.514 | 0.506 | 0.536 | **0.543** |
| DE-HR | 0.300 | 0.245 | 0.240 | 0.318 | 0.329 | 0.324 | 0.328 | 0.359 | **0.360** |
| DE-IT | 0.493 | 0.496 | 0.447 | 0.506 | 0.510 | 0.508 | 0.510 | 0.529 | **0.533** |
| DE-RU | 0.322 | 0.272 | 0.245 | 0.411 | 0.425 | 0.413 | 0.423 | **0.458** | 0.449 |
| DE-TR | 0.253 | 0.237 | 0.215 | 0.280 | 0.284 | 0.278 | 0.284 | **0.324** | 0.321 |
| EN-DE | 0.521 | 0.520 | 0.486 | 0.542 | 0.544 | 0.532 | 0.545 | 0.580 | **0.590** |
| EN-FI | 0.292 | 0.000 | 0.262 | 0.383 | 0.396 | 0.380 | 0.396 | 0.438 | **0.439** |
| EN-FR | 0.626 | 0.632 | 0.613 | 0.652 | 0.654 | 0.642 | 0.649 | 0.675 | **0.679** |
| EN-HR | 0.268 | 0.000 | 0.000 | 0.325 | 0.336 | 0.336 | 0.334 | 0.375 | **0.382** |
| EN-IT | 0.600 | 0.608 | 0.577 | 0.624 | 0.625 | 0.612 | 0.625 | **0.652** | 0.649 |
| EN-RU | 0.323 | 0.000 | 0.259 | 0.454 | 0.464 | 0.449 | 0.467 | **0.510** | 0.502 |
| EN-TR | 0.288 | 0.294 | 0.000 | 0.327 | 0.335 | 0.328 | 0.335 | 0.386 | **0.407** |
| FI-FR | 0.368 | 0.348 | 0.000 | 0.362 | 0.362 | 0.350 | 0.351 | 0.395 | **0.407** |
| FI-HR | 0.280 | 0.228 | 0.208 | 0.288 | 0.294 | 0.293 | 0.294 | 0.321 | **0.335** |
| FI-IT | 0.355 | 0.000 | 0.263 | 0.353 | 0.355 | 0.348 | 0.356 | 0.388 | **0.407** |
| FI-RU | 0.312 | 0.001 | 0.231 | 0.340 | 0.342 | 0.327 | 0.342 | 0.376 | **0.379** |
| HR-FR | 0.402 | 0.000 | 0.282 | 0.372 | 0.374 | 0.365 | 0.364 | 0.412 | **0.426** |
| HR-IT | 0.389 | 0.000 | 0.045 | 0.366 | 0.364 | 0.368 | 0.366 | 0.399 | **0.415** |
| HR-RU | 0.376 | 0.000 | 0.309 | 0.367 | 0.372 | 0.365 | 0.374 | 0.404 | **0.408** |
| IT-FR | 0.667 | 0.662 | 0.629 | 0.668 | 0.669 | 0.664 | 0.665 | 0.682 | **0.684** |
| RU-FR | 0.463 | 0.005 | 0.000 | 0.469 | 0.470 | 0.478 | 0.466 | 0.494 | **0.497** |
| RU-IT | 0.463 | 0.450 | 0.394 | 0.474 | 0.474 | 0.476 | 0.475 | 0.491 | **0.503** |
| TR-FI | 0.246 | 0.000 | 0.173 | 0.260 | 0.269 | 0.270 | 0.268 | 0.300 | **0.306** |
| TR-FR | 0.341 | 0.000 | 0.000 | 0.337 | 0.338 | 0.333 | 0.333 | 0.375 | **0.380** |
| TR-HR | 0.223 | 0.133 | 0.138 | 0.250 | 0.259 | 0.244 | 0.255 | 0.285 | **0.288** |
| TR-IT | 0.332 | 0.000 | 0.243 | 0.331 | 0.335 | 0.330 | 0.336 | 0.368 | **0.372** |
| TR-RU | 0.200 | 0.000 | 0.119 | 0.285 | 0.290 | 0.262 | 0.289 | **0.324** | 0.319 |
| Avg. | **0.375** | 0.183 | 0.253 | 0.400 | 0.405 | 0.398 | 0.403 | 0.437 | **0.442** |

Table 8: Mean Average Precision (MAP) of alignment methods on XLING with 5K supervision dictionaries, retrieval method is nearest neighbors. Benchmark results obtained from Glavaš et al. (2019) in which (*) Proc-B was reported using a 3K seed dictionary and (‡) PROC was evaluated without preprocessing and RCSLS was evaluated with Centering + Length Normalization (C+L) preprocessing.

## B EFFECTS OF RUNNING MULTIPLE ITERATIONS OF FIPP OPTIMIZATION

Although other alignment approaches (Joulin et al., 2018b; Artetxe et al., 2018c) run multiple iterations of their alignment objective, we find that running multiple iterations of the FIPP Optimization does not improve performance. We run between 1 and 5 iterations of the FIPP objective. For 1K seed dictionaries, 26/28 language pairs perform best with 1 iteration while 26/28 language pairs perform best with 1 iteration for 5K seed dictionaries. In the case of 1K seed dictionaries, (EN, FI) with 2 iterations and (RU, FR) with 2 iterations resulted in MAP performance increases of 0.002 and 0.001. For 5K seed dictionaries, (EN, FI) with 3 iterations and (EN, FR) with 2 iterations resulted in MAP performance increases of 0.002 and 0.002. Due to limited set of language pairs on which performance improvements were achieved, these results have not been included in Tables 1 and 6.

## C SELF-LEARNING FRAMEWORK

A self-learning framework, used for augmenting the number of available training pairs without direct supervision, has been found to be effective (Glavaš et al., 2019; Artetxe et al., 2018c) both in the case of small seed dictionaries or in an unsupervised setting. We detail a self-learning framework which improves the BLI performance of FIPP in the XLING 1K setting but not in the case of 5K seed pairs.

| Cos. Sim. Rank | English word | French word |
|:---:|:---:|:---:|
| 1 | oversee | superviser |
| 2 | renegotiation | renégociation |
| 3 | inform | informer |
| 4 | buy | acheter |
| 5 | optimize | optimiser |
| 6 | participate | participer |
| 7 | conceptualization | conceptualisation |
| 8 | internationalization | internationalisation |
| 9 | renegotiate | renégocier |
| 10 | interactivity | interactivité |

Table 9: Top 10 pairings by cosine similarity when using a Self-Learning framework on the English-French language pair.

Let $X_s \in \mathbb{R}^{c \times d_1}$ and $X_t \in \mathbb{R}^{c \times d_2}$ be the source and target embeddings for pairs in the seed dictionary $D$. Additionally, $\mathfrak{X}_s \in \mathbb{R}^{n \times d_1}$ and $\mathfrak{X}_t \in \mathbb{R}^{m \times d_2}$ are the source and target embeddings for the entire vocabulary. Assume all vectors have been normalized to have an $\ell_2$ norm of 1. Each source vector $\mathfrak{X}_{s,i}$ and target vector $\mathfrak{X}_{t,j}$ can be rewritten as $d$ dimensional row vectors of inner products with their corresponding seed dictionaries: $A_{s,i} = \mathfrak{X}_{s,i} X_s^T \in \mathbb{R}^{1 \times c}$ and $A_{s,j} = \mathfrak{X}_{t,j} X_s^T \in \mathbb{R}^{1 \times c}$.

For each source word $i$, we compute the target word $j$ with the greatest cosine similarity, equivalent to the inner product for vectors with norm of 1, in this $d$ dimensional space as $(i, j) = \arg\max_j A_{s,i} A_{t,j}^T$. For XLING BLI 1K experiments, we find the 14K $(i, j)$ pairs with the largest cosine similarity and augment our seed dictionaries with these pairs. In Table 7, the top 10 translation pairings, sorted by cosine similarity, obtained using this self-learning framework are shown for the English to French (EN-FR) language pair.

## D    COMPARISON OF FIPP SOLUTION TO ORTHOGONAL ALIGNMENTS

In this section, we conduct experimentation to quantify the degree to which the FIPP alignment differs from an orthogonal solution and compare performance on monolingual tasks before and after FIPP alignment.

### D.1    DEVIATION OF FIPP FROM THE CLOSEST ORTHOGONAL SOLUTION

For each language pair, we quantify the deviation of FIPP from an Orthogonal solution by first calculating the FIPP alignment before rotation, $\tilde{X}_s$, on the seed dictionary. We then compute the relative deviation of the FIPP alignment with the closest orthogonal alignment on the original embedding, $X_s$. This is equal to $D = \frac{\|\tilde{X}_s - \Omega^* X_s\|_F}{\|X_s\|_F}$ where $\Omega^* = \arg\min_{\Omega \in O(d_2)} \|X_s \Omega - \tilde{X}_s\|_F$. The average of these deviations for 1K seed dictionaries is 0.292 and for 5K seed dictionaries is 0.115. Additionally, the 3 language pairs with the largest and smallest deviations from orthogonal solutions are presented in the Table below. We find that in most cases, small deviations from orthogonal solutions are observed between languages in the same language family (i.e. Indo-European -> Indo-European) while those in different Language families tend to have larger deviations (i.e. Turkic -> Indo-European). A notable exception to this observation is English and Finnish which belong to different language families, Indo-European and Uralic respectively, yet have small deviations in their FIPP solution compared to an orthogonal alignment.

| Lang. Pair (1K) | $\ell_2$ Deviation | Lang. Pair (5K) | $\ell_2$ Deviation |
|:---|:---:|:---|:---:|
| German-Italian (DE-IT) | 0.025 | English-Finnish (EN-FI) | 0.008 |
| English-Finnish (EN-FI) | 0.090 | Croatian-Italian (HR-IT) | 0.012 |
| Italian-French (IT-FR) | 0.100 | Croatian-Russian (HR-RU) | 0.014 |
| Turkish-French (TR-FR) | 0.405 | Finnish-French (FI-FR) | 0.343 |
| Turkish-Russian (TR-RU) | 0.408 | Finnish-Croatian (FI-HR) | 0.351 |
| Turkish-Finnish (TR-FI) | 0.494 | Turkish-Finnish (TR-FI) | 0.384 |

Table 10: Smallest and Largest Deviations of FIPP from Orthogonal Solution, XLING BLI 1K and 5K

### D.2 EFFECT OF INNER PRODUCT FILTERING ON WORD-LEVEL ALIGNMENT

In FIPP, Inner Product Filtering is used to find common geometric information by comparing pairwise distances between a source and target language. To illustrate this step with translation word pairs, in the Table below we show the 5 words with the largest and smallest fraction of zeros, i.e. the "least and most filtered", in the binary filtering matrix $\Delta$ during alignment between English (En) and Italian (It). The words which are least filtered tend to have an individual word sense, i.e. proper nouns, while those which are most filtered are somewhat ambiguous translations. For instance, while the English word "securing" can be translated to the Italian word "fissagio", depending on the context the Italian words "garantire", "assicurare" or "fissare" may be more appropriate.

| Least Filtered | | Most Filtered | |
|---|---|---|---|
| English Word | Italian Word | English Word | Italian Word |
| japanese | giapponese | securing | fissagio |
| los | los | case | astuccio |
| china | cina | serves | servi |
| piano | pianoforte | joining | accoppiamento |
| film | film | fraction | frazione |

Table 11: Most and Least Filtered word pairs during FIPP's Inner Product Filtering for English-Italian alignment

### D.3 MONOLINGUAL TASK PERFORMANCE OF ALIGNED EMBEDDINGS

As FIPP does not perform an orthogonal transform, it modifies the inner products of word vectors in the source embedding which can impact performance on monolingual task accuracy. We evaluate the aligned embedding learned using FIPP, $\tilde{\mathfrak{X}}_s$, on monolingual word analogy tasks and compare these results to the original fastText embeddings $\mathfrak{X}_s$. In Table 3, we compare monolingual English word analogy results for English embeddings $\tilde{\mathfrak{X}}_s$ which have been aligned to a Turkish target embedding using FIPP. Evaluation of the aligned and original source embeddings on multiple English word analogy experiments show that aligned FIPP embeddings retain performance on monolingual analogy tasks.

Table 12: Monolingual Analogy Task Performance for English embedding before/after alignment to Turkish embedding.

| English Word Analogy Similarity | | |
|---|---|---|
| **Dataset** | Original | FIPP |
| WS-353 | **0.739** | 0.736 |
| MTurk-771 | 0.669 | **0.670** |
| SEMEVAL-17 | **0.722** | **0.722** |
| SIMLEX-999 | **0.382** | 0.381 |

## E EFFECT OF HYPERPARAMETERS ON BLI PERFORMANCE

In our experiments, we tune the hyperparameters $\epsilon$ and $\lambda$ which signify the level of discrimination in the inner product filtering step and the weight of the transfer loss respectively. When tuning, we account for the sparsity of $\Delta^\epsilon$ by scaling $\lambda$ in the transfer loss by $\gamma = \frac{c^2}{NNZ(\Delta^\epsilon)}$ where $NNZ(\Delta^\epsilon)$ is the number of nonzeroes in $\Delta^\epsilon$. Values of $\epsilon$ used in tuning were $[0.01, 0.025, 0.05, 0.10, 0.15]$ and scaled values of $\lambda$ used in tuning were $[0.25, 0.5, 0.75, 1.0, 1.25]$. Values of $(\epsilon, \lambda)$ which are close to one another result in similar performance and results are deterministic across reruns of the same experiment. As no validation set is provided, hyperparameters are tuned by holding out 20% of the training set.

# F    DIFFERENCE IN SOLUTIONS OBTAINED USING LOW RANK APPROXIMATIONS AND SGD

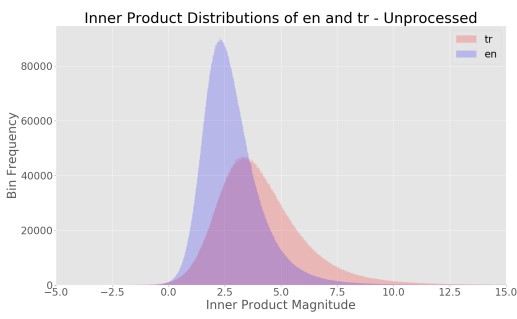

Figure 3: Comparison of FIPP Objective Loss for (FI-FR) for solutions obtained using SGD vs LRA

As detailed in Section 3, solutions to FIPP can be calculated either using Low Rank Approximations or Stochastic Gradient Descent (SGD). In this section, we show the error on the FIPP objective for SGD trained over 10,000 epochs on alignment of a Finnish (FI) embedding to a French (FR) embedding. The Adam (Kingma & Ba, 2015) optimizer is used with a learning rate of $1e^{-3}$ and the variable being optimized $\tilde{X}_s^{SGD}$ is initialized to the original Finnish embedding $X_s$. We find that the SGD solution $\tilde{X}_s^{SGD}$ approaches the error of the Low Rank Approximation $\tilde{X}_s^{LRA}$, which is the global minima of the FIPP objective, as shown in the Figure below but is not equivalent. While the deviation between SGD and the Low Rank Approximation, $D_{SGD,LRA} = \frac{\|\tilde{X}_s^{SGD} - \tilde{X}_s^{LRA}\|_F}{\|\tilde{X}_s^{LRA}\|_F} = 0.041$,
is smaller than the deviation between the original embedding and the Low Rank Approximation, $D_{Orig,LRA} = \frac{\|X_s - \tilde{X}_s^{LRA}\|_F}{\|\tilde{X}_s^{LRA}\|_F} = 0.343$ we note that the solutions are close but not equivalent.

# G    EFFECTS OF PREPROCESSING ON INNER PRODUCT DISTRIBUTIONS

We plot the distributions of inner products, entries of $X_s X_s^T$ and $X_t X_t^T$, for English (En) and Turkish (Tr) words in the XLING 5K training dictionary before and after preprocessing in Figure below. All embeddings used in experimentation are fastText word vectors trained on Wikipedia (Bojanowski et al., 2017). Since inner products between $X_s$ and $X_t$ are compared directly, the isotropic preprocessing utilized in FIPP is necessary for removing biases caused by variations in scaling, shifts, and point density across embedding spaces.

Figure 4: Gram matrix entries - unprocessed fast-Text embeddings

Figure 5: Gram matrix entries - fastText embeddings with preprocessing

# H    COMPLEXITY ANALYSIS

In this section, we provide the computational and space complexity of the FIPP method as described in the paper for computing $\tilde{X}_s$. We split the complexity for each step in the method. We leave out steps (i.e. preprocessing) which do not contribute significantly to the runtime or memory footprint. The computational complexity of matrix multiplication between two matrices $A_1 A_2$, where $A_1 \in \mathbb{R}^{m \times n} A_2 \in \mathbb{R}^{n \times p}$, is denoted as $MM(m,n,p)$ which is upper bounded by $2mnp$ operations.

### H.1 FIPP OPTIMIZATION

The complexity for solving the FIPP objective is detailed below:

$$\text{Space Complexity} = \underbrace{4c^2}_{\Delta^\epsilon,\, G^s,\, G^t,\, \tilde{G}^s} + \underbrace{3cd_2}_{X_s,\, X_t,\, \tilde{X}_s}$$

$$\text{Time Complexity} = \underbrace{\mathcal{O}(d_2 c^2)}_{\substack{\text{Compute } \tilde{X}_s \tilde{X}_s^T \\ \text{w/ Power Iteration}}} + \underbrace{3c^2}_{\|\tilde{X}_s \tilde{X}_s^T - X_s X_s^T\|_F} + \underbrace{5c^2}_{\lambda\|\Delta^\epsilon \circ (\tilde{X}_s \tilde{X}_s^T - X_t X_t^T)\|_F} = \mathcal{O}(d_2 c^2) \quad (8)$$

### H.2 SVD ALIGNMENT AND LEAST SQUARE PROJECTION

The complexity of the alignment using the Orthogonal Procrustes solution and the Least Squares Projection is as follows:

$$\text{Space Complexity} = \underbrace{3c^2}_{U\Sigma V^T} + \underbrace{cn}_{S} + \underbrace{nd_2}_{\tilde{\mathfrak{x}}_s^T}$$

$$\text{Time Complexity} = \underbrace{MM(c, d_2, d_2) + MM(d_2, d_2, d_2)}_{\tilde{X}_s V U^T} + \underbrace{d_2^3}_{SVD(X_t^T \tilde{X}_s)}$$

$$+ \underbrace{MM(c, d_1, n) + MM(c, d_2, c) + c^3 + MM(d_2, c, n)}_{\text{Least Squares}} \quad (9)$$

$$\leq 2(cd_2^2 + \frac{3}{2} d_2^3 + cd_1 n + c^2 d_2 + cd_2 n) + c^3$$

### H.3 DISCUSSION

In performing our analysis, we note that the majority of operations performed are quadratic in the training set size $c$. While we incur a time complexity of $\mathcal{O}(c^3)$ during our Least Squares Projection due to the matrix inversion in the normal equation, this inversion is a one time cost. The space complexity of FIPP is $\mathcal{O}(c^2)$ which is tractable as $c$ is at most $5K$. Empirically, FIPP is fast to compute taking less than $30$ seconds for a seed dictionary of size $5K$ which is more efficient than competing methods.

## I  RELATED WORKS: UNSUPERVISED ALIGNMENT METHODS

Cao et al. (2016) studied aligning the first two moments of sets of embeddings under Gaussian distribution assumptions. In Zhang et al. (2017), an alignment between embeddings of different languages is found by matching distributions using an adversarial autoencoder with an orthogonal regularizer. Artetxe et al. (2017) proposes an alignment approach which jointly bootstraps a small seed dictionary and learns an alignment in a self-learning framework. Hoshen & Wolf (2018) first projects embeddings to a subspace spanned by the top $p$ principle components and then learns an alignment by matching the distributions of the projected embeddings. Grave et al. (2019) proposes an unsupervised variant to the Orthogonal Procrustes alignment which jointly learns an orthogonal transform $\Omega \in \mathbb{R}^{d \times d}$ and an assignment matrix $P \in \{0, 1\}^{n \times n}$ to minimize the Wasserstein distance between the embeddings subject to an unknown rotation. Three approaches utilize the Gram matrices of embeddings in computing alignment initializations and matchings. Artetxe et al. (2018b) studied the alignment of dissimilar language pairs using a Gram matrix based initialization and robust self-learning. Alvarez-Melis & Jaakkola (2018) proposed an Optimal Transport based approach using the Gromov-Wasserstein distance, $GW(C, C', p, q) = \min_{\Gamma \in \Pi(p,q)} \sum_{i,j,k,l} L(C_{ik}, C'_{jl}) \Gamma_{ij} \Gamma_{kl}$ where $C, C'$ are Gram matrices for normalized embeddings and $\Gamma$ is an assignment. Aldarmaki et al. (2018) learns a unsupervised linear mapping between a source and target language with a loss at each iteration equal to the sum of squares of proposed source and target Gram matrices.

# J  ABLATION STUDIES

## J.1  AGGREGATE MAP PERFORMANCE FOR BLI TASKS

We provide an ablation study to quantify improvements associated with the three modifications of our alignment approach compared to a standard Procrustes alignment: (i) isotropic pre-processing (IP), (ii) inner product filtering (IPF) and (iii) weighted procrustes objectives (WP), on the XLING 1K and 5K BLI tasks. For seed dictionaries of size 1K, improvements associated with each portion of FIPP are approximately the same while for seed dictionaries of size 5K,

| Dict | 1K | % Imprv. | 5K | % Imprv. |
|---|---|---|---|---|
| Procrustes | 0.299 | - | 0.405 | - |
| FIPP w/o IP | 0.333 | 11.4% | 0.430 | 6.2% |
| FIPP w/o IPF | 0.335 | 12.0% | 0.439 | 8.4% |
| FIPP w/o WP | 0.336 | 12.4% | 0.440 | 8.6% |
| FIPP | 0.344 | 15.1% | 0.442 | 9.1% |
| FIPP + SL (+14K) | 0.406 | 35.8% | 0.441 | 8.9% |

Table 13: FIPP Ablation study of Mean Average Precision (MAP) on XLING 1K and 5K BLI task.

a larger improvement is obtained due to isotropic pre-processing than inner product filtering or weighted procrustes alignment.

## J.2  PREPROCESSING ABLATION

We perform an ablation study to understand the impact of preprocessing on XLING 1K and 5K BLI performance both on Procrustes and FIPP. Additionally, we compare the isotropic preprocessing used in our previous experimentation with iterative normalization, a well performing preprocessing method proposed by Zhang et al. (2017).

For training dictionaries of size 1K, iterative normalization and isotropic preprocessing before running procrustes result in equivalent aggregate MAP performance of 0.316. We find that iterative normalization and isotropic preprocessing each achieve better performance on 11 of 28 and 12 of 28 language pairs respectively.

Utilizing isotropic preprocessing before running FIPP results higher aggregate MAP performance for 1K training dictionaries (0.344) when compared to iterative normalization (0.331). In this setting, isotropic preprocessing achieves better performance on all 28 language pairs.

| Method | Proc. + IN | Proc. + IP | FIPP + IN | FIPP + IP |
|---|---|---|---|---|
| DE-FI | **0.278** | 0.264 | 0.291 | **0.296** |
| DE-FR | 0.421 | **0.422** | 0.438 | **0.463** |
| DE-HR | **0.240** | 0.239 | 0.264 | **0.268** |
| DE-IT | 0.452 | **0.458** | 0.468 | **0.482** |
| DE-RU | **0.336** | 0.331 | 0.304 | **0.359** |
| DE-TR | **0.191** | 0.182 | 0.212 | **0.215** |
| EN-DE | 0.484 | **0.490** | 0.490 | **0.513** |
| EN-FI | **0.301** | 0.299 | 0.304 | **0.314** |
| EN-FR | 0.578 | **0.585** | 0.582 | **0.601** |
| EN-HR | 0.250 | **0.258** | 0.261 | **0.275** |
| EN-IT | 0.562 | **0.563** | 0.575 | **0.591** |
| EN-RU | 0.372 | **0.375** | 0.381 | **0.399** |
| EN-TR | 0.262 | **0.267** | 0.275 | **0.292** |
| FI-FR | **0.245** | **0.245** | 0.256 | **0.274** |
| FI-HR | **0.214** | **0.214** | 0.239 | **0.243** |
| FI-IT | 0.270 | **0.272** | 0.287 | **0.309** |
| FI-RU | 0.249 | **0.251** | 0.274 | **0.285** |
| HR-FR | **0.255** | 0.251 | 0.270 | **0.283** |
| HR-IT | **0.273** | **0.273** | 0.296 | **0.318** |
| HR-RU | **0.285** | 0.279 | 0.310 | **0.318** |

| | | | | |
|---|---|---|---|---|
| IT-FR | **0.612** | **0.612** | 0.622 | **0.639** |
| RU-FR | **0.362** | 0.359 | 0.374 | **0.383** |
| RU-IT | 0.384 | **0.386** | 0.400 | **0.413** |
| TR-FI | **0.181** | 0.177 | 0.193 | **0.200** |
| TR-FR | **0.226** | **0.226** | 0.235 | **0.251** |
| TR-HR | **0.155** | 0.154 | 0.169 | **0.184** |
| TR-IT | **0.232** | 0.230 | 0.247 | **0.263** |
| TR-RU | 0.173 | **0.177** | 0.191 | **0.205** |
| AVG | **0.316** | **0.316** | 0.331 | **0.344** |

Table 14: Mean Average Precision (MAP) of alignment methods on XLING with 1K supervision dictionaries using either Iterative Normalization (IN) (Zhang et al., 2017) or Isotropic Preprocessing (IP).

For training dictionaries of size 5K, iterative normalization and isotropic preprocessing before running procrustes result in comparable aggregate MAP performances of 0.422 and 0.424 respectively. We find that iterative normalization and isotropic preprocessing each achieve better performance on 8 of 28 and 16 of 28 language pairs respectively.

Utilizing isotropic preprocessing before running FIPP results higher aggregate MAP performance for 5K training dictionaries (0.442) when compared to iterative normalization (0.425). In this setting, isotropic preprocessing achieves better performance on all 28 language pairs.

| Method | Proc. + IN | Proc. + IP | FIPP + IN | FIPP + IP |
|---|---|---|---|---|
| DE-FI | **0.378** | 0.370 | 0.383 | **0.389** |
| DE-FR | 0.519 | **0.526** | 0.521 | **0.543** |
| DE-HR | **0.345** | 0.344 | 0.349 | **0.360** |
| DE-IT | **0.523** | **0.523** | 0.522 | **0.533** |
| DE-RU | **0.432** | 0.428 | 0.435 | **0.449** |
| DE-TR | **0.309** | 0.300 | 0.314 | **0.321** |
| EN-DE | 0.562 | **0.574** | 0.562 | **0.590** |
| EN-FI | 0.420 | **0.424** | 0.425 | **0.439** |
| EN-FR | 0.660 | **0.667** | 0.660 | **0.679** |
| EN-HR | 0.361 | **0.370** | 0.360 | **0.382** |
| EN-IT | 0.641 | **0.642** | 0.643 | **0.649** |
| EN-RU | 0.483 | **0.489** | 0.482 | **0.502** |
| EN-TR | **0.370** | **0.370** | 0.368 | **0.407** |
| FI-FR | 0.377 | **0.383** | 0.386 | **0.407** |
| FI-HR | 0.315 | **0.316** | 0.318 | **0.335** |
| FI-IT | **0.382** | **0.382** | 0.386 | **0.407** |
| FI-RU | **0.362** | 0.361 | 0.366 | **0.379** |
| HR-FR | 0.397 | **0.400** | 0.402 | **0.426** |
| HR-IT | 0.396 | **0.397** | 0.396 | **0.415** |
| HR-RU | **0.394** | 0.390 | 0.397 | **0.408** |
| IT-FR | 0.671 | **0.674** | 0.671 | **0.684** |
| RU-FR | 0.481 | **0.482** | 0.485 | **0.497** |
| RU-IT | 0.488 | **0.489** | 0.490 | **0.503** |
| TR-FI | **0.288** | 0.286 | 0.292 | **0.306** |
| TR-FR | 0.352 | **0.355** | 0.354 | **0.380** |
| TR-HR | 0.267 | **0.272** | 0.274 | **0.288** |
| TR-IT | **0.353** | **0.353** | 0.354 | **0.372** |
| TR-RU | **0.304** | 0.303 | 0.304 | **0.319** |
| AVG | 0.422 | **0.424** | 0.425 | **0.442** |

Table 15: Mean Average Precision (MAP) of alignment methods on XLING with 5K supervision dictionaries using either Iterative Normalization (IN) (Zhang et al., 2017) or Isotropic Preprocessing (IP).

## J.3 WEIGHTED PROCRUSTES ABLATION

In order to measure the effect of a Weighted Procrustes rotation on BLI performance, we perform an ablation on both the 1K and 5K XLING BLI datasets against the standard Procrustes formulation.

For training dictionaries of size 1K, weighted procrustes achieves a marginally better aggregate MAP performance when compared to standard procrustes - 0.344 vs 0.336 respectively. In 27 of 28 language pairs, weighted procrustes provides improved MAP performance over standard procrustes.

With training dictionaries of size 5K, weighted procrustes achieves a marginally better aggregate MAP performance when compared to standard procrustes - 0.442 vs 0.440 respectively. In 21 of 28 language pairs, weighted procrustes provides improved MAP performance over standard procrustes.

| Method | FIPP + P (1K) | FIPP + WP (1K) | FIPP + P (5K) | FIPP + WP (5K) |
|--------|---------------|----------------|---------------|----------------|
| DE-FI | 0.286 | **0.296** | 0.386 | **0.389** |
| DE-FR | 0.446 | **0.463** | 0.541 | **0.543** |
| DE-HR | 0.260 | **0.268** | 0.358 | **0.360** |
| DE-IT | 0.471 | **0.482** | **0.534** | 0.533 |
| DE-RU | 0.350 | **0.359** | 0.444 | **0.449** |
| DE-TR | 0.207 | **0.215** | **0.321** | **0.321** |
| EN-DE | 0.508 | **0.513** | 0.589 | **0.590** |
| EN-FI | **0.314** | **0.314** | **0.440** | 0.439 |
| EN-FR | 0.594 | **0.601** | 0.678 | **0.679** |
| EN-HR | 0.271 | **0.275** | **0.382** | **0.382** |
| EN-IT | 0.581 | **0.591** | 0.648 | **0.649** |
| EN-RU | 0.393 | **0.399** | **0.503** | 0.502 |
| EN-TR | 0.285 | **0.292** | 0.405 | **0.407** |
| FI-FR | 0.268 | **0.274** | 0.403 | **0.407** |
| FI-HR | 0.233 | **0.243** | 0.330 | **0.335** |
| FI-IT | 0.294 | **0.309** | 0.402 | **0.407** |
| FI-RU | 0.274 | **0.285** | 0.376 | **0.379** |
| HR-FR | 0.276 | **0.283** | 0.424 | **0.426** |
| HR-IT | 0.305 | **0.318** | 0.413 | **0.415** |
| HR-RU | 0.307 | **0.318** | 0.405 | **0.408** |
| IT-FR | 0.628 | **0.639** | 0.683 | **0.684** |
| RU-FR | 0.381 | **0.383** | **0.498** | 0.497 |
| RU-IT | 0.400 | **0.413** | 0.500 | **0.503** |
| TR-FI | 0.194 | **0.200** | 0.303 | **0.306** |
| TR-FR | 0.249 | **0.251** | 0.379 | **0.380** |
| TR-HR | 0.169 | **0.184** | 0.284 | **0.288** |
| TR-IT | 0.253 | **0.263** | **0.372** | **0.372** |
| TR-RU | 0.195 | **0.205** | 0.317 | **0.319** |
| AVG | 0.336 | **0.344** | 0.440 | **0.442** |

Table 16: Mean Average Precision (MAP) of FIPP on XLING 5K and 1K supervision dictionaries using with either weighted procrustes (WP) or procrustes (P) rotation.

## K    BLI PERFORMANCE WITH ALTERNATIVE RETRIEVAL CRITERIA

In order to measure the effect on BLI performance of different retrieval criteria, we perform experimentation using CSLS and Nearest Neighbors on both the 1K and 5K XLING BLI datasets.

We find that the CSLS retrieval criterion provides significant performance improvements, compared to Nearest Neighbors search, both for Procrustes and FIPP on both 1K and 5K training dictionaries. For 1K training dictionaries, CSLS improves aggregate MAP, compared to Nearest Neighbors search, by 0.051 and 0.043 for Procrustes and FIPP respectively. In the case of 5K training dictionaries, CSLS improves aggregate MAP, compared to Nearest Neighbors search, by 0.049 and 0.031 for Procrustes and FIPP respectively.

| Method | Proc. + NN (1K) | Proc. + CSLS (1K) | FIPP + NN (1K) | FIPP + CSLS (1K) |
|---|---|---|---|---|
| DE-FI | 0.264 | **0.329** | 0.296 | **0.358** |
| DE-FR | 0.428 | **0.474** | 0.463 | **0.509** |
| DE-HR | 0.225 | **0.278** | 0.268 | **0.317** |
| DE-IT | 0.421 | **0.499** | 0.482 | **0.526** |
| DE-RU | 0.323 | **0.379** | 0.359 | **0.407** |
| DE-TR | 0.169 | **0.220** | 0.215 | **0.260** |
| EN-DE | 0.458 | **0.525** | 0.513 | **0.555** |
| EN-FI | 0.271 | **0.336** | 0.314 | **0.365** |
| EN-FR | 0.579 | **0.623** | 0.601 | **0.641** |
| EN-HR | 0.225 | **0.280** | 0.275 | **0.314** |
| EN-IT | 0.535 | **0.599** | 0.591 | **0.637** |
| EN-RU | 0.352 | **0.398** | 0.399 | **0.427** |
| EN-TR | 0.225 | **0.278** | 0.292 | **0.328** |
| FI-FR | 0.239 | **0.285** | 0.274 | **0.332** |
| FI-HR | 0.187 | **0.233** | 0.243 | **0.279** |
| FI-IT | 0.247 | **0.310** | 0.309 | **0.358** |
| FI-RU | 0.233 | **0.271** | 0.285 | **0.317** |
| HR-FR | 0.248 | **0.279** | 0.283 | **0.331** |
| HR-IT | 0.247 | **0.317** | 0.318 | **0.364** |
| HR-RU | 0.269 | **0.313** | 0.318 | **0.354** |
| IT-FR | 0.615 | **0.647** | 0.639 | **0.675** |
| RU-FR | 0.352 | **0.393** | 0.383 | **0.429** |
| RU-IT | 0.360 | **0.418** | 0.413 | **0.462** |
| TR-FI | 0.169 | **0.206** | 0.200 | **0.242** |
| TR-FR | 0.215 | **0.254** | 0.251 | **0.303** |
| TR-HR | 0.148 | **0.179** | 0.184 | **0.214** |
| TR-IT | 0.211 | **0.269** | 0.263 | **0.306** |
| TR-RU | 0.168 | **0.201** | 0.205 | **0.237** |
| AVG | 0.299 | **0.350** | 0.344 | **0.387** |

Table 17: Mean Average Precision (MAP) of FIPP and Procrustes on XLING with 1K supervision dictionaries using with either (CSLS) or Nearest Neighbors (P) retrieval criteria.

| Method | Proc. + NN (5K) | Proc. + CSLS (5K) | FIPP + NN (5K) | FIPP + CSLS (5K) |
|---|---|---|---|---|
| DE-FI | 0.359 | **0.433** | 0.389 | **0.440** |
| DE-FR | 0.511 | **0.560** | 0.543 | **0.576** |
| DE-HR | 0.329 | **0.386** | 0.360 | **0.400** |
| DE-IT | 0.510 | **0.555** | 0.533 | **0.568** |
| DE-RU | 0.425 | **0.456** | 0.449 | **0.461** |
| DE-TR | 0.284 | **0.334** | 0.321 | **0.357** |
| EN-DE | 0.544 | **0.584** | 0.590 | **0.608** |

| | | | | |
|---|---|---|---|---|
| EN-FI | 0.396 | **0.453** | 0.439 | **0.479** |
| EN-FR | 0.654 | **0.686** | 0.679 | **0.696** |
| EN-HR | 0.336 | **0.394** | 0.382 | **0.422** |
| EN-IT | 0.625 | **0.665** | 0.649 | **0.674** |
| EN-RU | 0.464 | **0.499** | 0.502 | **0.517** |
| EN-TR | 0.335 | **0.390** | 0.407 | **0.429** |
| FI-FR | 0.362 | **0.417** | 0.407 | **0.447** |
| FI-HR | 0.294 | **0.348** | 0.335 | **0.374** |
| FI-IT | 0.355 | **0.427** | 0.407 | **0.442** |
| FI-RU | 0.342 | **0.392** | 0.379 | **0.405** |
| HR-FR | 0.374 | **0.434** | 0.426 | **0.461** |
| HR-IT | 0.364 | **0.427** | 0.415 | **0.449** |
| HR-RU | 0.372 | **0.430** | 0.408 | **0.438** |
| IT-FR | 0.669 | **0.698** | 0.684 | **0.703** |
| RU-FR | 0.470 | **0.522** | 0.497 | **0.537** |
| RU-IT | 0.474 | **0.517** | 0.503 | **0.529** |
| TR-FI | 0.269 | **0.326** | 0.306 | **0.346** |
| TR-FR | 0.338 | **0.391** | 0.380 | **0.413** |
| TR-HR | 0.259 | **0.310** | 0.288 | **0.329** |
| TR-IT | 0.335 | **0.389** | 0.372 | **0.408** |
| TR-RU | 0.290 | **0.323** | 0.319 | **0.338** |
| AVG | 0.405 | **0.454** | 0.442 | **0.473** |

Table 18: Mean Average Precision (MAP) of FIPP and Procrustes on XLING with 5K supervision dictionaries using with either (CSLS) or Nearest Neighbors (P) retrieval criteria.

