# OpenReview forum: "Filtered Inner Product Projection for Crosslingual Embedding Alignment"
_ICLR.cc/2021/Conference — ICLR 2021 Poster_

### Official Review · AnonReviewer1 · 2020-10-28
**An interesting method, but the paper needs additional insights, experiments, and analyses (REVISED)**

**Rating:** 6
**Confidence:** 4

**Review:**

=== After the revision and the author response ===
I would like to thank the authors for their very elaborate response. I acknowledge that I might have asked for too many experiments in my review, but this was mostly because I really wanted to understand the various aspects of the method (as one other reviewer mentions - it has too many 'moving parts') and ensure that all the comparisons are valid by comparing to the latest work. While it is not possible (due to time and computational restrictions) to run all the required experiments, I am still happy to raise my score. While the paper might not be so impactful, it might still be a pretty nice and quick-to-compute BLI baseline for any future developments in this area.
=== ===

This paper presents a new method to learn projection/mapping-based cross-lingual word embeddings, termed FIPP (Filtered Inner Product Projection), which shows some promising results in supervised settings with 5K dictionaries. The paper is clearly written and easy to follow, and the main results across three tasks (bilingual lexicon induction + 2 downstream tasks) following a standard evaluation setup demonstrate that the method can often outperform its competitors. Another advantage of the method is the efficiency of learning the alignment; the method is several orders of magnitude quicker than standard choices such as VecMap or RCSLS. Although the paper does provide a valuable contribution to the field of learning (static) cross-lingual word embeddings, I still have quite some concerns and additional questions related to this work, and I am not ready to accept this work in its current format.

*The authors mention (even in the abstract) that FIPP is applicable "even when the source and target embeddings are of differing dimensionalities." However, I don't see how previous approaches are not applicable in these setups - the linear or non-linear mapping can be formulated between embedding spaces of different dimensionalities in a general case. Therefore, I don't see why this is important to stress with FIPP. This is some inaccurate argumentation imo.

*The reconstruction loss obtained from the Gram matrices has not been used in bilingual settings, but this idea is directly borrowed from prior work on monolingual embeddings, so I do not see it as a methodological contribution. The second term of FIPP, however, reminds me of a recent work on instance-based mapping and learning local mapping functions, as proposed and validated by Glavas and Vulic (ACL 2020) and Nakashole et al. (several papers). However, the authors do not discuss this very relevant recent work nor provide any comparisons to these approaches, which seem closely related.
-- I would also suggest the authors to include some other very relevant recent models in their comparison. For instance, what about the following papers:
- https://arxiv.org/abs/2004.13889 (Mohiuddin et al.)
- https://doi.org/10.1162/coli_a_00374 (Mohiuddin and Joty, Computational Linguistics 2020)
- https://doi.org/10.18653/v1/p19-1018 (Patra et al., ACL 2019)
- https://www.aclweb.org/anthology/2020.acl-main.766/ (Anastasopoulos et al., ACL 2020)
- https://openreview.net/forum?id=S1l-C0NtwS (Wang et al., ICLR 2020)
- There are other recent papers...

This field is a rapidly evolving field, and comparing to the models which were SotA in 2019 is definitely not sufficient, especially if the good results are supposed to be the main contribution of the paper.

*Along the same line as the previous point, preprocessing of word embeddings can be quite crucial to the final performance, I wonder if the authors have also tried a well-performing preprocessing strategy from Zhang et al. (ACL 2019), also with the baselines. Would this change the ranking of the models in the comparison?

*Concerning the previous point, the paper lacks a comprehensive ablation study. Given that it does blend several different components (e.g., multi-loss, preprocessing regime, dictionary size), it is quite underwhelming to see that the authors did not conduct any experiments to further understand what components of the model are more crucial to the final performance, and how different choices of hparams affect the performance. Further, there are no experiments with different dictionary sizes and we're left wondering if the method would work with a smaller number of training examples (e.g., 500 examples or 1K examples as evaluated by Vulic et al., EMNLP 2019). What about typologically more dissimilar languages, etc.? What about self-learning strategies? Is it possible to improve the results with FIPP by running several iterations of the method? Have the authors tried that? How does preprocessing affect the final performance? What about the hyper-parameter epsilon? How is that hyperparameter related to language (dis)similarity?

-- In sum, a complete piece of work should provide answers to all these questions, and it is still quite difficult to judge whether this particular model will really advance the induction of cross-lingual word embeddings. Does it also rely on the approximate isomorphism assumption (Soogard et al., ACL 2018) and how does language distance relate to the transfer loss?

*Notes on motivation: While working on static cross-lingual word embeddings is still an important research thread, the paper should provide a thorough discussion on why these models are still important in the context of the current mainstay of multilingual/cross-lingual NLP based on massively multilingual models such as multilingual BERT, XLM-R, multilingual T5, etc. An open question is what applications (besides bilingual dictionary induction) could still benefit from static cross-lingual word embeddings.

Minor comments:
- Mikolov et al. did not propose using the Orthogonal Procrustes solution per se - they solved the problem via SGD and did not impose any orthogonality constraint (which is needed to solve it analitically via the Procrustes method - see later work of Xing et al., and the survey paper of Ruder et al.)

- The title is not exactly accurate: 'multilingual embedding alignment' would suppose aligning more than 2 languages (3 and more), while the authors, as most of prior work, learn only bilingual word embeddings. Another question concerns the possibility to extend this idea to shared spaces with more than 2 languages. How feasible is this?

---

> ### Author Response · Authors · 2020-11-22
> **Response to Reviewer 1**
>
> Dear Reviewer 1,
>
> We really appreciate your valuable feedback.
>
> **First, we want to summarize the main contributions of our paper:**
>
> (1) We propose FIPP, a supervised approach for aligning a source and target embedding to a common representation space using pairwise inner products.
>
> (2) Unlike previous approaches, FIPP aligns a source and target embedding to equal rank vector spaces even when their dimensionalities are different.
>
> (3) FIPP is more efficient than competing approaches, often providing an order of magnitude speedup without any specialized hardware.
>
> (4) FIPP provides better performance than previous approaches on the XLING (5K) dataset and also, when utilizing a self-learning framework, the XLING (1K) dataset.
>
> (5) FIPP performs robustly on downstream evaluations.
>
> (6) Additionally, we propose a weighted variant of Procrustes which takes into consideration in-sample error of supervision translation pairs and improves BLI performance for most language pairs.
>
> **Here are the replies to each of your question/comment:**
>
> **1. For the question "The authors mention (even in the abstract) that FIPP is applicable ... This is some inaccurate argumentation imo."**
>
> Thank you for bringing up this point, we have changed the messaging to be more precise throughout the paper. While it is feasible to modify certain methods, such as Procrustes to a linear mapping as shown in Section 6 of the main paper, to align embeddings of different dimensionalities, previous methods are not able to align source and target embeddings to equivalent rank vector spaces ("isomorphic vector spaces" not the same as the graph isomorphism mentioned in (Soogard et al., ACL 2018)). We have added further analysis on this point to Section 6.2 of the main paper. To understand why this is a concern, consider the case where a lower dimensional embedding is being aligned to higher dimensional embedding (i.e. $d_1 < d_2$). Other methods, which are capable of being formulated to this setting, align a $d_1$ dimensional vector space to the $d_2$ dimensional vector space in $\mathbb{R}^{d_2}$ by "padding" on a null space. This is undesirable as the two embeddings, while aligned, have different ranks and the null space of the aligned embedding will be $\mathbb{R}^{d_2 - d_1}$. This can lead to issues when learning and transferring models on the shared representation space. First, rank-deficient features can lead to instability in optimization (Boyd and Vanderberghe, 2004) of models using the source but not the target embedding. This may be remedied with regularization. Even still, regularized models transferred from the source space to the target space will be learned on a rank of $d_1$ and not be able to use all of the column features from the target embedding. Meanwhile, models transferred from the target space to the source space will have features associated with the null space of the source embedding which is also undesirable as it leads to biased models. To highlight this point, we align an English Embedding $X_s \in \mathbb{R}^{c \times d_1}$ and a German Embedding $X_t \in \mathbb{R}^{c \times d_2}$ where $d_1 = 200$ and $d_2 = 300$ using a linear mapping, RCSLS and FIPP. The spectrums of the aligned embedding's Gram matrices are plotted in Section 6.2 of the main paper. Additionally, it should be noted that many approaches impose orthogonality constraints, even those which do not necessarily produce a structure preserving mapping (i.e. GeoMM), which are not feasible when dimensionalities differ.

---

> > ### Author Response · Authors · 2020-11-22
> > **Continued Response to Reviewer 1**
> >
> > **2. For the question "The reconstruction loss obtained from the Gram matrices has not ... are supposed to be the main contribution of the paper.**
> >
> > Thank you for pointing out the recent work by Glavas and Vulic (ACL 2020) which indeed is most similar to our method and has strong results on dissimilar language pairs. We have added this to the related works,  section 2.2 of our main paper. This method also utilizes a non-linear mapping for source and target vectors; for each word it learns an individual "translation vector" using a weighted average of the cosine similarities of the nearest $D$ words in a language's embedding space multiplied by the euclidean distances to the corresponding translations of these nearest neighbors. FIPP learns a non-linear mapping to make modifications to a source embedding's Gram matrix using differences in inner products on a filtered set of source and target word pairs. The aligned embeddings are then obtained using an Eigendecomposition of this modified Gram matrix. While there are similarities with between these works, we note that there a functional differences in the selection and weighting of pairs from which to make modifications, what is being modified in a non-linear manner (i.e. source Gram Matrix and resulting Eigenvalues/Eigenvectors vs explicit source and target embeddings), and approaches for solving alignments. Additionally the efficiency of these approaches are quite different, after running Instamap (https://github.com/codogogo/instamap) on the 5K DE-TR dictionary, we noticed that the runtime is 1860s compared to 24s for FIPP.
> >
> > For the Nakashole papers, we are assuming that you are referencing (Nakashole and Flauger, ACL 2018) and (Nakashole, EMNLP 2018) as were the most recent works by the author on this topic that we could find. (Nakashole and Flauger, ACL 2018) presents interesting analysis which finding that locally linear maps vary between different neighborhoods in bilingual embedding spaces which suggests that global alignments are nonlinear in structure. (Nakashole, EMNLP 2018) presents an unique method for embedding alignment using neighborhood sensitive maps which shows strong performance on dissimilar language pairs. We have added both these papers to our Related Works in Section 2.2 of the main paper. These works show similarities to FIPP in their intuition but not in their methodological contributions; in terms of evaluation, we have been unable to find code for these methods on the author's website, aclweb.org or github.
> >
> > We agree that there are numerous innovative contributions on this research topic which have been introduced recently and have shown strong performance on BLI tasks. One point we would like to mention about evaluation is that almost all of these works primarily operate on the MUSE dataset which has been shown to be somewhat misleading (Kementchedjhieva et. al., ACL 2019) and utilize the Precision @ 1 metric which is less informative than MAP (Glavas et. al. ACL 2019).
> >
> > Wang et al., ICLR 2020 is a complementary work to ours as it is a framework used in conjunction with an alignment method such as RCSLS. Mohiuddin and Joty, Computational Linguistics 2020 is a well performing unsupervised method however the authors do not compare against well performing supervised approaches, only against Procrustes. Anastasopoulos et al., ACL 2020 is focused on the related task of finding "hub" languages for inducing mappings for more than 2 languages. The other recent papers provide strong performance in BLI tasks. Mohiuddin et al. EMNLP 2020 achieves SOTA performance on low-resource MUSE language pairs and Patra et al., ACL 2019 outperforms previous approaches on well-studied MUSE language pairs. Due to the runtime considerations, hyperparameter budgets, and the number of experiments, 28 language pairs with 2 different dictionary sizes, we have been unable to provide an BLI evaluation for the methods from these two papers during the response period but have added them to Related Works section 2.2 and can include evaluations on these methods to the final version of our paper. Additionally, we have included related unsupervised methods which utilize inner product information to Section K of the Appendix.

---

> > > ### Author Response · Authors · 2020-11-22
> > > **Continued Response to Reviewer 1**
> > >
> > > **3. For the question "Along the same line as the previous point, preprocessing of word embeddings can be quite crucial ... How does preprocessing affect the final performance?"**
> > >
> > > This is a great question also asked by Reviewer 4. We have performed ablation studies for both the isotropic preprocessing and the iterative normalization preprocessing proposed by (Zhang et al. ACL 2019) for FIPP and Procrustes in Section J.2 of the Appendix for all 28 language pairs on both the 1K (w/o self-learning) and 5K supervision dictionaries. We find that both preprocessing approaches yield similar performance improvements in Procrustes. When both procrustes and FIPP are compared with their best preprocessings, FIPP approx. provides on the 1K dictionaries (w/o self-learning) a 10\% relative improvement and on the 5K dictionaries a 5\% relative improvement.
> > >
> > > In terms of other models in comparison (i.e. RCSLS), in Zhang et al. (ACL 2019) the improvement for RCSLS from preprocessing was shown to be much smaller than for Procrustes. This is due to RCSLS already using Centering + Length Normalization (one iteration of Iterative Normalization) in it's default form. For language pairs also contained in our BLI experiments, the following improvements for RCSLS were shown in Zhang et al. (ACL 2019) for the P@1 metric- EN-DE: +0.6\%, EN-FI: +1.0\%, EN-FR: +0.0\%, EN-HR: +0.5\%, EN-IT: +0.3\%, EN-RU: +0.8\%, EN-TR: +1.7\%. We performed experimentation on RCSLS, using the default hyperparameters specified in the FastText repo,  using the standard C+L preprocessing and the Iterative Normalization (IN) implementation from Zhang et al. (ACL 2019) from (https://gist.github.com/zhangmozhi/1e37c997514115e9b63476e322ca2ad0). For the 1K seed dictionaries, IN resulted in an improvement in MAP of +0.001, a +0.3\% relative improvement from 0.320 $->$ 0.321, and for the 5K seed dictionaries an improvement in MAP of +0.002, a +0.5\% relative improvement from 0.431 $->$ 0.433.
> > >
> > > Since the 112 BLI experiments required to run this ablation, for a single set of RCSLS hyperparameters, took close to 15 hours to run we unfortunately cannot report performance with a sufficient amount of tuning during the response period. However, using Iterative Normalization, RCSLS may provide comparable performance to FIPP on the 5K dictionaries but likely not on the 1K dictionaries (esp. when using self-learning). We have made a note in our reporting both in Section 5 of the main paper and Section A of the Appendix that Procrustes performance is without preprocessing and that RCSLS is with C+L normalization.
> > >
> > >
> > > **4. For the question "Concerning the previous point, the paper lacks a comprehensive ablation study. Given that it does blend ... and how different choices of hparams affect the performance."**
> > >
> > > Thank you for this suggestion, we have updated the paper to include ablations for:
> > >
> > > - Aggregate BLI performance with the removal of each individual component in FIPP: Section J.1 of the Appendix
> > > - Preprocessing Approaches (both Isotropic Preprocessing and Iterative Normalization): Section J.2 of the Appendix
> > > - Weighted Procrustes vs. Procrustes: Section J.3 of the Appendix
> > >
> > > **5. For the question "Further, there are no experiments with different dictionary sizes ... training examples (e.g., 500 examples or 1K examples as evaluated by Vulic et al., EMNLP 2019)."**
> > >
> > > Experimentation on both 1K and 5K dictionaries for all 28 language pairs in XLING is provided in Section A of the Appendix, which has been moved to follow the bibliography rather than in a separate file in the supplementary material. With the addition of a self-learning approach, which your review prompted us to consider, FIPP outperforms other surveyed methods on the 1K dataset ($>7$\% relative performance improvement to Proc-B (Glavas et. al., ACL 2019)). Also, we have modified Section 5.2 of the main paper to include 10 sample language pairs each for 1K and 5K dictionaries.
> > >
> > > **6. For the question "What about typologically more dissimilar languages, etc.?"**
> > >
> > > While the languages used in our experimentation comprise 3 language families (Indo-European, Turkic, and Uralic) there are some notable ones that we have missed (i.e. Sino-Tibetan, Afroasiatic, ..) . We agree that it would be useful to evaluate on more distant languages and will do so in future work.

---

> > > > ### Author Response · Authors · 2020-11-22
> > > > **Continued Response to Reviewer 1**
> > > >
> > > > **7. For the question "What about self-learning strategies?"**
> > > >
> > > > Thank you for this suggestion, a self-learning strategy significantly improves BLI performance of FIPP, to the best amongst compared methods, in the case of 1K seed dictionaries (MRR: 0.344 $->$ 0.406). We detail the results in Appendix Section A and provide the self-learning approach we used along with some sample augmentation word pairs for English-French in Section C of the Appendix. It should be noted that self-learning results in a 10x increase in runtime (24s $->$ 200s) and the requirement of a GPU; additionally, self-learning does not affect performance in the case of 5K seed dictionaries.
> > > >
> > > > **8. For the question "Is it possible to improve the results with FIPP by running several iterations of the method? Have the authors tried that?"**
> > > >
> > > > For the majority of language pairs, running FIPP for multiple iterations does not improve performance, at least in our experiments which ran up to 5 iterations. We detail the few cases in which multiple iterations results in performance improvements in Section B of the Appendix.
> > > >
> > > > **9. For the question "What about the hyper-parameter epsilon? How is that hyperparameter related to language (dis)similarity? Does it also rely on the approximate isomorphism assumption (Soogard et al., ACL 2018) and how does language distance relate to the transfer loss?"**
> > > >
> > > > We find that language dissimilarity is not explicitly related with the hyper-parameter epsilon but rather with the magnitude of the transfer loss, which is determined by both epsilon and lambda, and controls the degree to which the resulting alignment differ from an orthogonal solution. This is measured by finding the normed difference between the FIPP solution and the closest orthogonal solution and detailed along with examples in Appendix Section D.1. FIPP does not rely on the approximate graph isomorphism assumption from (Soogard et al., ACL 2018) as the geometric structure of the source vector spaces, and therefore the nearest neighbors in these spaces, are not necessarily preserved during alignment as shown in Appendix Section D.1.
> > > >
> > > >
> > > > **10. For the question "Notes on motivation: While working on static cross-lingual word embeddings ...  could still benefit from static cross-lingual word embeddings."**
> > > >
> > > > Thank you for pointing this out. We have updated the motivation in the first paragraph of Section 1 of the main paper. The main points of motivation for alignment of static cross-lingual word embeddings that we discuss are (1) Improvements to models using static cross-lingual embeddings, (2) Enabling the study of linguistic patterns across languages, (3) Recent work (Artexte et. al. ACL 2020) which has shown performance improvements in certain tasks, such as multilingual document classification, but not others such as crosslingual natural language inference, when using static cross-lingual alignment approaches (i.e. VecMap) with contextual embedding models (i.e. m-BERT). Although, there may be additional applications of these alignment methods in different problem settings than those we have included, as you've mentioned, this is still an open question.
> > > >
> > > > **11. For the question "Mikolov et al. did not propose using the Orthogonal Procrustes solution per se ... and the survey paper of Ruder et al.)."**
> > > >
> > > > Thank you for pointing this out, this was also mentioned by Reviewer 4. (Xing et. al., HLT 2015) proposed solving the sum of squares objective using an SQP and (Artexte et. al., ACL 2017) and (Smith et. al., ICLR 2017) independently showed that procrustes provides a closed-form solution for the sum of squares objective with an orthogonality constraint. We have corrected the attributions for the orthogonal procrustes solution in Section 1 and 2.2 of the main paper.
> > > >
> > > > **12. For the question "The title is not exactly accurate: ... How feasible is this?"**
> > > >
> > > > We appreciate this suggestion; the title has been changed to '... Crosslingual embedding alignment' as this is the most commonly used term in preceding literature. In terms of extensions to shared spaces with more than 2 languages, previous work such has shown that bilingual approaches can been used in multilingual settings by individually aligning multiple languages to a individual "pivot" or "hub" language (Ammar et. al. 2016, https://arxiv.org/pdf/1602.01925v2.pdf; Smith et. al. 2017,  https://github.com/babylonhealth/fastText_multilingual). This type of multilingual extension should be feasible using FIPP without modifications to the method. Additionally, approaches such as GeoMM (Jawanpuria et. al., TACL 2019) have explicit multilingual formulations which are shown to better align in the multilingual setting. While it seems possible to extend the transfer loss in the FIPP objective to include terms for each additional language in the multilingual setting, we have not evaluated in this problem setting.

---

### Official Review · AnonReviewer3 · 2020-10-29
**An interesting, effective and efficient method for Bilingual Lexicon Induction**

**Rating:** 8
**Confidence:** 4

**Review:**

**Summary:**
This paper proposes a new approach for Bilingual Lexicon Induction (BLI). To do this, the paper proposes the use of a FIPP objective, which consists of the linear combination of reconstruction and transfer loss terms. It derives the Gram matrix that minimizes this loss, followed by a rank-constrained semi-definite approximation to obtain an aligned embedding whose Gram matrix is close to the one minimizing the FIPP. The proposed approach is deterministic and has been shown to be efficient. Experiments also demonstrate the method works well, both on BLI and on downstream tasks.

**Strengths:**
* The proposed method is interesting and elegant, and novel in the context of BLI. FIPP also performs well empirically, demonstrated both by its superior performance on BLI (as shown on with an evaluation on 28 XLing pairs) and on its comparable performance in the downstream CLDC and XNLI tasks. While the proposed approach does not do the best on the presented downstream tasks, it seems to perform well (reasonably comparably to the best) consistently, which none of the other methods seem to do.
* FIPP allows for BLI using embeddings of different dimensions, and performs well even when the embeddings are of different dimensions, as shown experimentally for en-de. To the best of my knowledge, no other method supports this.
* The proposed method is quite efficient because it can be represented as a few matrix operations. This has been demonstrated by comparing its runtimes with two other SotA approaches (VecMap and RCSLS).

**Comments/Questions:**
* One drawback is that the proposed method does not perform as well for the 1k case (Appendix). Some analysis about why that might be the case would have been nice. That being said, the same is the case for the other supervised methods as well (PROC-B aside). I also appreciate the authors' transparency in this regard.
* The authors present the downstream performance on 2 (CLDC, XNLI) of the 3 tasks (the other being CLIR) used in [5]. Is there a reason these 2 specific tasks were chosen, but not CLIR?
* The proposed method uses a weighted Procrustes objective to rotate $\widetilde{X}_s$, which is a minor detail, but whose use is, as far as I am aware, novel in the context of BLI. An ablation here to show how much the weighted Procrustes objective improves over the regular Procrustes transform (such as MUSE [4] does) would have been nice to see (Eg: a row "FIPP w/o WP w/ P" in Appendix Table 7).
* GeoMM [3] is very efficiently optimizable as well. Do the authors have an estimate about how FIPP performs compared to GeoMM, with respect to the average time they each take on CPU?
* The Appendix had some nice experimental details, additional experiments and insights. It would be good to add the Appendix to the end of the main paper (as opposed to separately) if possible.

**Minor points:**
* For XNLI, [1] should probably be cited along with [2], since XNLI was introduced by [1] based on the English-only MultiNLI corpus introduced by [2]
* Section 6.2 didn't seem to flow very well with the rest of Section 6 (which follows Section 5 nicely). Consider moving it to before Section 5 if possible (I'm not entirely certain where it would fit best, possibly as a new Section 3.3).

[1] Conneau, Alexis, et al. "XNLI: Evaluating cross-lingual sentence representations." arXiv preprint arXiv:1809.05053 (2018).
[2] Williams, Adina, Nikita Nangia, and Samuel R. Bowman. "A broad-coverage challenge corpus for sentence understanding through inference." arXiv preprint arXiv:1704.05426 (2017).
[3] Jawanpuria, Pratik, et al. "Learning multilingual word embeddings in latent metric space: a geometric approach." Transactions of the Association for Computational Linguistics 7 (2019): 107-120.
[4] Conneau, Alexis, et al. "Word translation without parallel data." arXiv preprint arXiv:1710.04087 (2017).
[5] Glavas, Goran, et al. "How to (properly) evaluate cross-lingual word embeddings: On strong baselines, comparative analyses, and some misconceptions." arXiv preprint arXiv:1902.00508 (2019).



============================================================

**Update:**

A few  suggestions on the latest version:

* Adding pointers from Table 1 and 2 to their full equivalents in the Appendix would be helpful.
* The Appendix ablations are now thorough and exhaustive, but parsing them and digesting what they represent is a little tricky. Adding a one-line summary (much like in Section 5.`1) would be very helpful here. For example, talking about in how many pairs FIPP+IP outperforms Proc.+IP,  FIPP+IN outperforms Proc.+IN, FIPP+IP outperforms FIPP+IN.

and a few comments:

* The sheer number of experiments that the authors have performed in general (and in particular in the relatively short time period of this rebuttal) is impressive, and is in my experience indicative of an extremely good, well-designed and easy to iterate upon framework/code-base. If my guess is correct, I urge the authors to release their code if possible, because I believe it would greatly help anyone working in this space (or even consuming BLI's output in a downstream task).
* It is also very heartening to see how much R1's suggestion helps improve the 1k case! However, because the efficiency and the lack of need of a GPU are big selling points, the drawback of adding self-supervision (10x the time, need for a GPU) might be a good caveat to add in Section 6.1 as opposed to keeping it till Appendix A.

Overall, I would like to thank the authors for their very detailed and thorough response,  and for taking into account so much of all the reviewers' feedback to make the paper clearer with much more comprehensive ablations. The paper, its techniques proposed and their performance and efficiency, and the detailed experiments it conducts will be helpful for both the field of BLI and other fields relying on it. In view of this, this paper is a clear accept in my opinion, and I raise my score from 7 to 8.

---

> ### Author Response · Authors · 2020-11-22
> **Response to Reviewer 3**
>
> Dear Reviewer 3,
>
> We really appreciate your valuable feedback.
>
> **First, we want to summarize the main contributions of our paper:**
>
>
> (1) We propose FIPP, a supervised approach for aligning a source and target embedding to a common representation space using pairwise inner products.
>
> (2) Unlike previous approaches, FIPP aligns a source and target embedding to equal rank vector spaces even when their dimensionalities are different.
>
> (3) FIPP is more efficient than competing approaches, often providing an order of magnitude speedup without any specialized hardware.
>
> (4) FIPP provides better performance than previous approaches on the XLING (5K) dataset and also, when utilizing a self-learning framework, the XLING (1K) dataset.
>
> (5) FIPP performs robustly on downstream evaluations.
>
> (6) Additionally, we propose a weighted variant of Procrustes which takes into consideration in-sample error of supervision translation pairs and improves BLI performance for most language pairs.
>
> **Here are the replies to each of your question/comment:**
>
> **1. For the question "One drawback is that the proposed method does not perform as well for the 1k case (Appendix) ... appreciate the authors' transparency in this regard."**
>
> We found this surprising since the best performing supervised method in the 1K case (Proc-B) was not as competitive in the 5K case. Based on a suggestion by Reviewer 1, we implemented a self-learning strategy described in detail in Section C of the Appendix, similar to that which is used by both of the two best performing models in the 1K case (Artexte et. al., ACL 2018) and (Glavas et. al., ACL 2019). We find that FIPP outperforms these methods (MAP: 0.344 $->$ 0.406) when utilizing a self-learning strategy to augment the seed dictionary before alignment. These results are in Section A of the Appendix. Analysis of sample augmentation word pairs derived from this self-learning approach for the English-French language pair are also included in Appendix Section C. It should be noted that some downsides to the self-learning approach are a 10x increase in runtime (24s $->$ 200s), although still faster than (Artexte et. al., ACL 2018) and (Glavas et. al., ACL 2019), and the requirement of a GPU. Additionally, self-learning does not affect performance in the case of 5K seed dictionaries.
>
> **2. For the question "The authors present the downstream performance on 2 CLDC, XNLI) of the 3 tasks (the other being CLIR) used in [5] .. 2 specific tasks were chosen but not CLIR?"**
>
> Thanks for bringing this up. Unfortunately, the CLIR dataset (http://catalog.elra.info/en-us/repository/browse/ELRA-E0008/) is not open-source and requires payment (€300 non-members, €150 members) for access so we chose to evaluate on the other two tasks which are openly available.
>
> **3. For the question "The proposed method uses a weighted Procrustes objective to rotate ...  would have been nice to see (Eg: a row "FIPP w/o WP w/ P" in Appendix Table 7)."**
>
> Due to page margin constraints, we have been unable to add the column to Appendix Table 7 or Table 8. However, we have added ablation studies for FIPP using both weighted procrustes (WP) and regular procrustes (P) for resolving rotational symmetries in Section J.3 of the Appendix for all 28 language pairs on both the 1K (w/o self-learning) and 5K supervision dictionaries. We find that weighted procrustes provides a much larger relative improvement on 1K dictionaries (+2.5\%) compared to 5K dictionaries (+0.5\%). Additionally, the improvements due to weighted procrustes are fairly consistent across language pairs since in 50/56 experiments WP performed better than P. Also, to the best of our knowledge, weighted variants of procrustes have not been used before in the context BLI.
>
> **4. For the question "GeoMM [3] is very efficiently optimizable ... with respect to the average time they each take on CPU?"**
>
> We ran the implementation of GeoMM from the author's repo
> (https://github.com/anoopkunchukuttan/geomm) which does not provide a CPU implementation and has a requirement of 2 GPUs. On the machine used for profiling in the paper we profiled the runtime for the script  ./geomm\_results.sh at approximately 190s per language pair. When using the 'optimized' variant GeoMM took 20s per language pair although the repo states 6.5s. This is either due to limitations of our hardware or our inclusion of time used for loading the embeddings, constructing training matrices, and saving the embeddings which we have included in all of our runtime reporting (even though they are not method specific). Aside from hardware specs, both methods are very efficient and take roughly the same amount of time on the 5K supervision dictionaries.

---

> > ### Author Response · Authors · 2020-11-22
> > **Continued Response to Reviewer 3**
> >
> > **5. For the question "The Appendix has some nice experimental details, additional experiments and insights ... (as opposed to separately) if possible."**
> >
> > Thank you for this suggestion, we have moved the Appendix to follow the main paper rather than in a separate supplementary file.
> >
> >
> > **6. For the question "TFor XNLI, [1] should probably be cited along with [2] ... based on the English-only MultiNLI corpus introduced by [2]."**
> >
> > Thanks for pointing this out, we have corrected the attributions in Section 5.2.2 of the main paper.
> >
> > **7. For the question "Section 6.2 didn't seem to flow  ... possibly as a new Section 3.3)."**
> >
> > We agree, the previous version's Section 6.2 did not flow as well with the rest of the discussion section and we've moved it to a new Section 3.3.

---

### Official Review · AnonReviewer4 · 2020-10-30
**Good results but a lot of moving parts**

**Rating:** 6
**Confidence:** 4

**Review:**

This paper proposes an approach to align word vector spaces based on a dictionary of pairs (e.g. translations) that retains "common geometric structure" (inner products within each vector space for a subset of word pairs). In essence the approach applies preprocessing to each vector space, finds a projection of one vector spaces to the (lower-dimensional) vector space minimising a combination of a reconstruction and transfer loss, and then aligns the two using weighted procrustes to resolve rotational symmetries.

The topic is of interest to the community and related papers on vector space alignment have been published at previous ICLRs. Essentially, this paper combines insights from a number of different works, with good results. However, overall the solution is relatively complex with a lot of moving parts.

Strengths:
* The paper is largely well written and easy to follow.
* The proposed method can be applied if vector spaces have differing dimensionality
* Results compare well to existing methods

Weaknesses:
* As the initial projection is not constrained to be orthogonal, the geometric structure within the projected vector space is altered (only relevant if dimensionality is the same). This is usually seen as a big advantage of procrustes-based solutions and not really discussed by the authors. How much does the solution found using this method deviate from an orthogonal solution?
* Introducing the isotropic preprocessing makes it difficult to compare the alignment procedures, as both word vector spaces are altered significantly (As shown in Arora where it significantly impacts performance on semantic similarity). The authors should explore the impact of this preprocessing also for other methods (e.g. procrustes).
* No comparison between weighted procrustes and regular procrustes to address the rotation symmetry for the inner product approach. Is it crucial to use the weighted variant here?
* Unclear how well this approach works in conjunction with retrieval strategies other than nearest neighbour - previous work found this to be a significant factor for performance.
* Only anecdotal evidence for comparison of SGD and low-rank approximation
* Related work: Mikolov 2013a did not propose a procrustes solution to the alignment problem. Instead, Artexte 2017 and Smith 2017 showed (independently) that procrustes provides an analytical solution to the optimisation problem proposed in Mikolov 2013 under the orthogonality constraint.

Smaller issues / suggestions:
* I found the "preview" of the technical approach in the introduction more confusing than clarifying, as some of the details mentioned later on where missing, e.g. that rows in $X_s$ and $X_t$ refer to the same word based on the dictionary.
* It would be great to get a more intuitive "feel" how the alignment that respects the "geometric structure" would be different to a procrustes alignment, e.g. by showing a few example translations
* The plural of corpus is corpora
* You assume dimensionality $d_2 < d_1$ but don't really state it anywhere?

Other (not considered in review):
* The CCA results are quite strong - If the authors are using the CCA implementation in scikit learn I would like to point out that it actually uses an iterative estimation of partial least squares, which works much better than a more naive implementation of CCA (as e.g. demonstrated in https://arxiv.org/abs/1905.05547)

---

> ### Author Response · Authors · 2020-11-22
> **Response to Reviewer 4**
>
> Dear Reviewer 4,
>
> We really appreciate your valuable feedback.
>
> **First, we want to summarize the main contributions of our paper:**
>
> (1) We propose FIPP, a supervised approach for aligning a source and target embedding to a common representation space using pairwise inner products.
>
> (2) Unlike previous approaches, FIPP aligns a source and target embedding to equal rank vector spaces even when their dimensionalities are different.
>
> (3) FIPP is more efficient than competing approaches, often providing an order of magnitude speedup without any specialized hardware.
>
> (4) FIPP provides better performance than previous approaches on the XLING (5K) dataset and also, when utilizing a self-learning framework, the XLING (1K) dataset.
>
> (5) FIPP performs robustly on downstream evaluations.
>
> (6) Additionally, we propose a weighted variant of Procrustes which takes into consideration in-sample error of supervision translation pairs and improves BLI performance for most language pairs.
>
> **Here are the replies to each of your question/comment:**
>
> **1. For the question "As the initial projection is not constrained to be orthogonal ... how much does the solution found using this method deviate from an orthogonal solution?"**
>
> We have added a discussion on the solutions obtained from FIPP and their deviations from orthogonal solutions in Section D of the Appendix. In particular, we find that the deviation of the FIPP solution with the closest orthogonal solution, $D = \frac{\|\tilde{X}_s - \Omega^* X_s\|_F}{\| X_s \|_F}, \Omega^* = argmin _{\Omega \in O(d_2)} || X_s \Omega - \tilde{X}_s  ||_F$, is heavily dependent on the language pair and, in particular, the similarity of the source and target languages. Typically, but not always, deviations are largest for languages in different language families (i.e. Turkish (Turkic) $->$ Finnish (Uralic), D = 0.384)  and smallest for those in the same family (i.e. Croatian (Indo-European) $->$ Italian (Indo-European), D = 0.012). As the geometric structure of the source embedding is not preserved by FIPP alignment, we performed monolingual analogy experimentation before and after alignment to a Turkish embedding (D = 0.151) and on this example language pair found that English monolingual performance was not detrimented by performing FIPP alignment as shown in Section D.3 of the Appendix.
>
> **2. For the question "Introducing the isotropic preprocessing makes it difficult ... The authors should explore the impact of this preprocessing also for other methods (e.g. procrustes)."**
>
> We have performed ablation studies for both the isotropic preprocessing and the iterative normalization preprocessing proposed by (Zhang et al. ACL 2019) for FIPP and Procrustes in Section J.2 of the Appendix for all 28 language pairs on both the 1K (w/o self-learning) and 5K supervision dictionaries. We find that both preprocessing approaches yield similar performance improvements in Procrustes. When both procrustes and FIPP are compared with their best preprocessings, FIPP approx. provides on the 1K dictionaries (w/o self-learning) a 10\% relative improvement and on the 5K dictionaries a 5\% relative improvement.
>
> **3. For the question "No comparison between weighted procrustes and regular procrustes ... Is it crucial to use the weighted variant here?"**
>
> We have added ablation studies for FIPP using both weighted procrustes (WP) and regular procrustes (P) for resolving rotational symmetries in Section J.3 of the Appendix for all 28 language pairs on both the 1K (w/o self-learning) and 5K supervision dictionaries. In particular, we find that the weighted procrustes provides a much larger relative improvement on 1K dictionaries (+2.5\%) compared to 5K dictionaries (+0.5\%). Additionally, the improvements due to weighted procrustes are fairly consistent across language pairs, in 50/56 experiments WP performed better than P.
>
> **4. For the question "Unclear how well this approach works in conjunction with retrieval strategies other than nearest neighbor..."**
>
> We have added experimentation comparing both FIPP and Procrustes using the CSLS (Conneau et. al., ICLR 2018) retrieval strategies to Section K of the Appendix. Using CSLS, both methods improve substantially in their BLI performance but we note that the performance gap between FIPP and Procrustes decreases. On the 1K seed dictionaries (w/o self-learning) the relative performance gap in MAP between FIPP and Procrustes drops from 15\% to 10.5\% and for the 5K seed dictionaries the gap in MAP between FIPP and Procrustes drops from 9.1\% to 4.2\%.

---

> > ### Author Response · Authors · 2020-11-22
> > **Continued Response to Reviewer 4**
> >
> > **5. For the question "Related work: Mikolov 2013a did not propose a ..."**
> >
> > Thank you for pointing out this error, the contributions of these works have been corrected in Section 1 and Section 2.2 of the main paper.
> >
> > **6. For the question "I found the 'preview' of the technical approach ... refer to the same word based on the dictionary."**
> >
> > We have clarified the technical approach "preview" in Section 1 of the main paper to include details which had been omitted but are critical for the problem definition.
> >
> > **7. For the question "It would be great to get a more intuitive 'feel' how the alignment ... e.g. by showing a few example translations."**
> >
> > In Section D.2, we provide some intuition for how the alignment respects "geometric structure" by highlighting words which are least filtered in the transfer loss and therefore contribute most to non-orthogonal modifications to the source embedding. In particular, for English-Italian, proper nouns and less-ambiguous terms such as "japanese" and "piano" constitute most to the transfer loss and are considered portions of the common geometric structure. Meanwhile, ambiguous verbs and polysemous nouns contribute least to the transfer loss. An example would be the English-Italian pair ("securing", "fissagio") which is a correct translation but the italian words garantire", "assicurare" or "fissare" may be more accurate translations depending on the context.
> >
> > **8. For the question "The plural of corpus is corpora".**
> >
> > We have corrected this error in Section 1 of the main paper.
> >
> > **9. For the question "You assume dimensionality $d_2 < d_1$ but don't really state it anywhere?".**
> >
> > In our experimentation in Section 6.2 we have assumed that $d_1 \leq d_2$, as this case is particularly difficult for performing alignment, but did not state this anywhere. This assumption has been added to Section 6.2.
> >
> > **10. For the question "The CCA results are quite strong ... works much better than a more naive implementation of CCA."**
> >
> > Thank you for pointing this out. Yes, the CCA implementation we are using is scikit-learn's \texttt{cross\_decomposition.CCA} which utilizes iterative estimation of PLS. We have noted this in Section 6.2.2 of the main paper. An additional point we have added is that for the experiments in Section 6.2.2 both the Linear and FIPP methods map both $X_s$, $X_t$ to $\mathbb{R}^{d_2}$ while CCA maps to $\mathbb{R}^{min(d_1, d_2)}$ which is a limitation that may be undesirable for instance if one seeks to transfer a model already learned on the target embedding in $\mathbb{R}^{d_2}$.

---

### Author Response · Authors · 2020-11-22
**Paper Revision**

**We thank the reviewers for their valuable feedback. Based on the questions and suggestions, we made the following content revisions in our paper which are of general interest:**

(1) We have moved the appendix to the end of the main paper instead of in a separate supplementary file. The appendix has been expanded to include more experimentation and analysis. In particular, numerous ablation studies, analysis of deviations of the FIPP solution to orthogonal solutions, and samples of linguistic analysis has been added.

(2) Based on the suggestion of Reviewer 1, we have added experimentation using a self-learning approach which augments the seed dictionary using cosine similarities between word pairs not included in the supervision set before performing alignment. This addition results in a significant performance improvement for the case of 1K seed dictionaries (MAP: 0.344 $->$ 0.406), better than that of previous methods. Due to time constraints, ablation studies for 1K seed dictionaries have been performed without self-learning but additional ablation studies on 1K w/ self-learning will be added to the final version of the paper

(3) We have changed the messaging around alignment of embeddings with differing dimensionality to be more precise. When the source embedding $X_s$ is being aligned to a target embedding $X_t$ of lower dimension ($d_1 > d_2$), an additional dimensionality reduction step, not required for FIPP, can be applied to other methods before performing alignment. However, in the case where the source embedding $X_s$ is being aligned to a target embedding $X_t$ ($d_1 < d_2$), other methods, to the best of our knowledge, are not able to perform alignment such that the aligned source embedding $\tilde{X}_s$ and the target embedding $X_t$ are equivalent rank vector spaces. As an example, for $d_1 < d_2$ a linear transform or RCSLS can align a $d_1$ dimensional embedding to a $d_2$ dimensional embedding by "padding" the aligned source embedding with a $\mathbb{R}^{d_2 - d_1}$ null space, which is undesirable in shared representation learning. In comparison, this is not done by FIPP which aligns embeddings to isomorphic vector spaces, analyzed in Section 6.2 of the main paper.

---

### Decision · Program_Chairs · 2021-01-07
**Final Decision**

**Decision:**

Accept (Poster)

**Comment:**

This paper proposes a method for bilingual lexicon induction.
The proposed method is efficient, it optimizes a reconstruction and transfer loss.
Extensive experiments are reported, and the methods provides improvements over prior work.
Overall, the paper brings together prior ideas in a useful way.